



# A comparison of modeled daytime E-regions from E-PROBED and PyIRI with ionosonde observations

Daniel J. Emmons[1], Cornelius Csar Jude H. Salinas[2], Dong L. Wu[2], Nimalan Swarnalingam[2,3], Eugene V. Dao[4], Jorge L. Chau[5], Yosuke Yamazaki[5], Kyle E. Fitch[1], and Victoriya V. Forsythe[6]

[1]Air Force Institute of Technology, Wright-Patterson AFB, OH, United States
[2]NASA Goddard Space Flight Center, Greenbelt, MD, United States
[3]The Catholic University of America, Washington, DC, United States
[4]Air Force Research Laboratory, Albuquerque, NM, United States
[5]Leibniz Institute for Atmospheric Physics, Kühlungsborn, Germany
[6]U.S. Naval Research Laboratory, Washington, DC, United States

**Correspondence:** Daniel J. Emmons (daniel.emmons@afit.edu)

**Abstract.** While the F-region is the primary focus of many ionospheric models because it contains the peak electron density, the E-region is an important region for ionospheric conductivities and high-frequency radio propagation. This study analyzes modeled E-regions from the newly developed PyIRI and E-PROBED models. A long-term comparison of E-region predictions from E-PROBED and PyIRI with ionosonde observations is performed for three sites spanning low- (Fortaleza, Brazil), mid- (El Arenosillo, Spain), and high-latitudes (Gakona, Alaska). Modeled $foE$ and $hmE$ trends are compared against a combination of manually-scaled and automatically-scaled ionograms using ARTIST-5 for the period 2009-2024 for El Arenosillo and Gakona, and 2015-2024 for Fortaleza. Measured and modeled virtual heights are compared for a subset of the ionograms through the use of a numerical ray-tracer. Overall, the models showed reasonable agreement with the ionosonde observations, with solar cycle, seasonal, and diurnal trends well captured for $foE$. E-PROBED generally overestimates $foE$ with Mean Absolute Relative Errors (MRAEs) peaking around 70% at dusk, while PyIRI showed close agreement with ionosonde $foE$ resulting in MRAE peaks around 10%. The $hmE$ predictions showed weaker agreement, with a 15-20 km overestimate from E-PROBED when compared against auto-scaled ionograms, and a constant $hmE$ prediction of 110 km for all times from PyIRI. However, manually-scaled $hmE$ estimates show close agreement with E-PROBED predictions, indicating that great care must be taken when using auto-scaled $hmE$. Modeled virtual heights derived from E-PROBED and PyIRI show reasonable agreement with ionosonde observations, providing confidence in altitude-integrated electron density profiles. A slight bias exists between the modeled and measured virtual heights, and the direction of the bias reverses for manual- versus auto-scaled ionograms, demonstrating that auto-scaled uncertainties are also present in the virtual height observations. Overall, these results indicate that E-PROBED and PyIRI provide reasonable E-region estimates and may be used for practical applications that require modeled E-region parameters.





## 1 Introduction

The E-region of the ionosphere plays an important role in ionospheric conductivities (Rishbeth and Garriott, 1969; Kelley, 2009) that impact ground-based magnetometer observations (Brekke et al., 1974; Yamazaki and Maute, 2017), atmospheric energy input and balance (Roble et al., 1987), and High Frequency (HF) radio propagation (Fabrizio, 2013). Therefore, a proper understanding of global E-region morphology can provide insight into both scientific and practical applications, especially

through the use and development of E-region models.

Critical frequencies of the E-region ($fo$E; or corresponding peak electron density $Nm$E) have been studied for many years, revealing a relationship with the solar zenith angle (Muggleton, 1972), season (Kouris and Muggleton, 1973b), sunspot number (Muggleton, 1971b), and Sun-Earth distance (Muggleton, 1971a). Global collections of ionosondes have provided insight into $fo$E variation over time, such that empirical relationships could be developed (e.g., Kouris and Muggleton (1973a); Kouris

(1998)). Similarly, rocket and incoherent scatter radar data have been used to calculate empirical $Nm$E trends (Chasovitin et al., 1985). As the E-region is photochemistry dominated and driven primarily by extreme ultraviolet (EUV) flux with wavelengths below 150 Å (Schunk and Nagy, 2009), chemistry models have been created (Titheridge, 1996, 1997), providing insight into difficult-to-measure densities (such as [NO]) and reaction rates and coefficients.

Models of the peak electron density altitude of the E-region, $hm$E, have also been developed (Ivanov-Kholodny et al., 1998;

Titheridge, 2000). These studies have shown that $hm$E remains nearly constant around local noon at lower altitudes, with increases in altitude near sunrise and sunset. The general behavior of $hm$E can be captured by Chapman theory (Chapman, 1931), providing a linear relationship with the natural log of the secant of the solar zenith angle. Titheridge (2000) derived a modified Chapman theory dependence for $hm$E taking into account season, latitude, solar flux, solar zenith angle and local time, resulting in $hm$E values between 105 and 120 km that agree well with ionosonde observations from Auckland, New

Zealand.

A Chapman-layer E-region can be approximated as a quasi-parabola (Bradley and Dudeney, 1973), requiring a peak altitude, peak density, and half-thickness (scale height) to describe the bottomside shape. These parameters can be calculated using virtual height measurements from ionosondes (Reinisch and Xueqin, 1983; Titheridge, 1985a), making ionosondes a powerful tool for extracting E-region electron density profiles (EDPs). For this reason, long-term ionosonde observations have been used

as the backbone of global bottomside E-region models such as the empirical International Reference Ionosphere (IRI; Bilitza (1990, 1998). While Incoherent Scatter Radar (ISR) observations contribute to IRI's estimates for the E-F valley, $fo$E and $hm$E are mainly driven by ionosonde observations (Bilitza et al., 2022). Recently, the core framework of IRI has been implemented in Python instead of the historical Fortran approach in the new PyIRI (Forsythe et al., 2024), allowing for more rapid execution of global ionospheric profiles. PyIRI uses the global coefficients used to drive IRI, while also implementing certain features of

NeQuick (Nava et al., 2008) for the top-side and E-region.

With recent improvements in Global Navigation Satellite System (GNSS) Radio Occultation (RO) techniques for extracting D- and E-region electron densities (Wu, 2018), global E-region observations are now available in regions that were previously unobtainable by ionosondes or ISR (Wu et al., 2022, 2023). These global GNSS-RO observations have been used to develop a



modern E-region model, E-region Prompt Radio Occultation Based Electron Density (E-PROBED; Salinas et al. (2024)). The model was developed using measurements from the Constellation Observing System for Meteorology, Ionosphere, and Climate (COSMIC-1) showing variability and asymmetry in E-region electron densities as a function of time, altitude, and latitude that are not predicted by empirical or physics-based models.

While the truly global spread of GNSS-RO observations provides great promise for remote sensing of the upper atmosphere, the integrated nature of the measurements (Hajj and Romans, 1998; Schreiner et al., 1999) motivates the need for additional comparison against more direct observations such as those implemented by ionosondes. The same argument holds for models derived from GNSS-RO (E-PROBED) versus ionosonde (PyIRI) observations of the E-region. A recent study by Shaver et al. (2023) has provided a framework for this comparison between EDPs derived from GNSS-RO, ionosondes, and models. In the present study, we implement many of the same approaches to compare modeled E-regions from E-PROBED and PyIRI to ionosonde observations used as the "ground-truth" validating dataset. This effort aims to provide insight into model performance as well as the solar cycle, seasonal, and diurnal morphology of the E-region for several ionosonde sites spanning low-, mid-, and high-latitudes.

## 2 Materials and Methods

Three Digisonde sites were selected as the basis for this comparison: Fortaleza, Brazil (URSI code FZA0M, 3.9° S, 321.6° E, -21.5° inclination), El Arenosillo, Spain (EA036, 37.1° N, 353.3° E, 50.6° inclination), and Gakona, USA (GA762, 62.4° N, 215.0° E, 75.5° inclination). These three sites span low-, mid-, and high-latitudes, providing a comparison over a variety of ionospheric conditions. Ionosonde virtual height observations, $foE$, and $hmE$ estimates were obtained from the Digital Ionogram Database (DIDBASE, 2025). Virtual heights were obtained using SAOExplorer version 3.6.1 (SAO-X, 2025) while the $foE$ and $hmE$ estimates were downloaded directly from DIDBASE. The virtual height observations have a frequency resolution of approximately 25 kHz and Digisondes use a standard temporal resolution of one ionogram every 15 minutes. However, the temporal resolution can be variable, sometimes increasing up to 5 minutes per ionogram. In addition, there are several periods with outages at each site, which can last anywhere from days to years. It should also be noted that the minimum $foE$ observations from ionosondes are constrained by the minimum transmit frequency and sensitivity, such that nighttime $foE$ values below ∼1.5 MHz are generally not measured by ionosondes. Therefore, the comparison performed here does not include nighttime observations or model estimates.

To provide a long-term comparison, automatically-scaled ionograms using the Automatic Real-Time Ionogram Scaler with True Height calculation version 5 (ARTIST-5; Galkin and Reinisch (2008)) were used for $foE$ and $hmE$ estimates. The start dates for implementing ARTIST-5 vary from site to site, with El Arenosillo implementing in December 2008, Fortaleza in November 2014, and Gakona in May 2007. From this, the $foE$ and $hmE$ comparison for each site begins on the date of ARTIST-5 implementation and continues through 2024. Within each of these periods, a collection of manually-scaled ionograms are also available, with the largest density for EA036 in 2009.



Although auto-scaled ARTIST-5 ionograms are known to differ from manually-scaled profiles (e.g., Stankov et al. (2023)), the use of auto-scaled ionograms allows for a prolonged comparison period to analyze long-term trends. In an attempt to remove poor quality ionograms from the comparison, ARTIST confidence scores were required to be above 90%. While it is not entirely clear how these confidence scores map to an uncertainty in electron density as a function of altitude, the 90% confidence threshold requires a series of quality control criteria to be satisfied such that noisy or problematic ionograms are removed (Galkin et al., 2013; Themens et al., 2022). Furthermore, we implemented additional criteria to be satisfied as an expanded quality control procedure: the $fo$E was required to be above the minimum transmitted frequency, fmin, profiles with sporadic-E ($fo$Es) observations were removed, $hm$E values were required to be above 90 km, and $hm$E estimates of exactly 110 km were removed. The removal of 110 km $hm$E values was implemented because of a large number of ionograms that defaulted to this altitude, likely following climatological estimates and not derived entirely from the observations. This 110 km $hm$E altitude will be discussed in more detail in Sections 3 and 4.

Geomagnetically quiet conditions were also enforced for the comparison by constraining Kp≤3, with Kp values obtained from NASA's OMNIWeb (Papitashvili and King, 2020). This ensures that differences in E-region observations and model predictions are not reliant on the model's ability to predict variations caused by geomagnetic activity. Both E-PROBED and IRI were run for each ionogram time satisfying the criteria outlined above. In total, this results in 19,727 $fo$E and $hm$E observations for EA036 (including 896 manually-scaled ionograms), 54,036 for FZA0M, and 3,837 for GA762.

For the models, E-PROBED version 1.0 (Salinas, 2024) was used to estimate E-region EDPs using longitude, latitude, altitude range (90-130 km with 0.25 km resolution), date, and time of day as input. The $fo$E and $hm$E estimates were calculated from the EDPs using Scipy's find_peaks function (Virtanen et al., 2020), that performed well on the smooth E-PROBED profiles with a single E-region peak. E-PROBED was developed from COSMIC-1 GNSS-RO observations of the E-region from 2007-2016 and is driven by Solar Zenith Angle (SZA), season, and F10.7 with an additional non-SZA component that is a function of latitude, local time, season, and F10.7 (Salinas et al., 2024). Time series of $Nm$E, $hm$E, and scale-height for each latitude-local time bin were then fit to Fourier coefficients up to the 10th harmonic, providing a lookup table for each bin that is called within E-PROBED.

PyIRI profiles were calculated using version 0.0.2 (Forsythe and Burrell, 2023) with an altitude range of 90-130 km and a resolution of 0.25 km. The PyIRI inputs are longitude, latitude, date, time of day, F10.7, altitude range, and an option for NmF2 calculations, ccir_or_ursi, which was set to use the CCIR coefficients. The $fo$E and $hm$E values were output directly from PyIRI.

Although $fo$E can be observed directly from virtual height observations as an E-region cusp (assuming $fo$E is outside of a restricted transmission band), $hm$E estimates are calculated through a quasi-parabolic fit to the virtual height data (e.g., Titheridge (1985b); Reinisch and Xueqin (1983)), which results in additional uncertainty for the $hm$E estimates. To account for this additional uncertainty in the altitude estimates, a comparison to directly measured ionogram virtual heights is performed on a subset of the data. Virtual height observations contain information on the shape and magnitude of EDPs through the group index of refraction (Budden, 1966), which provides a more direct comparison against ionosonde observations than comparing against inverted EDPs that require a series of assumptions on the profile shape, etc. (Shaver et al., 2023). The E-PROBED and




PyIRI EDPs were converted to virtual heights through the use of a High Frequency (HF) ray tracer. Specifically, the EPDs were input into Another Ionospheric Ray Tracer (AIRTracer; a model developed by Eugene V. Dao at the Air Force Research Laboratory) to calculate virtual heights for ordinary-mode rays. AIRTracer uses the Jones-Stephenson (Jones and Stephenson, 1975) formulation with the Booker quartic and no collisions, and has been rewritten in the Julia Programming Language to

decrease computation time. For each subset of ionograms used for the virtual height comparison, a group path is calculated for each transmit frequency of the ionogram virtual height data (roughly 25 kHz resolution) using the E-PROBED and PyIRI EDPs. The virtual height is then taken as half of the group path. Due to the additional processing time required to calculate the virtual heights of E-PROBED and PyIRI, the observation subset was limited to January-March 2009 for EA036 (total of 618 profiles), August 2019 for FZA0M (711 profiles), and May 2008 to January 2009 for Gakona (515 profiles). These periods were

selected when the data density was large for the site of interest, and the period selected for EA036 includes the manually-scaled ionograms. A total of at least 500 profiles was desired for each site, which resulted in variable time spans due to differences in ionogram cadence and quality (i.e., lack of ionospheric disturbances, etc.).

## 3 Results

The comparison results are separated by the ionosonde site, and the combined trends will be discussed in Section 4. For each

site, the trends are analyzed by year (solar cycle), day of year (seasonal), and solar local time (diurnal), with seasonal and diurnal results displayed in Appendix A. Then, the modeled virtual heights predicted by E-PROBED and PyIRI are compared with ionosonde observations for a subset of the ionograms.

### 3.1 El Arenosillo, Spain

The yearly $foE$ estimates for El Arenosillo, Spain (EA036) are shown in Figure 1, spanning from December 2008 to 2024.

Manually-scaled ionograms are marked by orange stars to distinguish from the auto-scaled ionograms, with the majority of manually-scaled ionograms taking place in 2009. For $foE$, the manually-scaled trends match the auto-scaled trends due to the E-region cusp in ionograms, which provide a direct feature to estimate $foE$ (e.g. Figure 1.3 of Piggott and Rawer (1961)).

Yearly quartiles are calculated for each dataset to show long-term trends. In Figure 1, the black trend lines intersect the yearly medians, and the 25% and 75% quartiles are shown as error bars for each year. The ionosonde observations show

a slight increase during Solar Cycle 24 with median $foE$ values of 3.1 MHz for 2014 and a range of 1.7–3.4 MHz. As mentioned previously, nighttime $foE$ observations fall below the minimum frequency measured by ionosondes, fmin, such that the minimum $foE$ observations do not include nighttime values. The median decreased to 2.5 MHz during solar minimum near 2020 with a sharp increase to 3.2 MHz in 2023 along with an extended range of 1.6–4.0 MHz. Seasonal trends are clearly visible with peaks in the local (boreal) summer, with an interesting double peak surrounding the summer of 2023 (upper right

of Figure 1). A more in-depth discussion of seasonal and diurnal trends is provided in Appendix A.

E-PROBED and PyIRI show a more pronounced solar cycle trend around solar maximum in 2014. The median $foE$ in 2014 for PyIRI is 3.5 MHz with a range of 1.8–4.2 MHz. Similarly, E-PROBED predicted a median $foE$ of 4.1 MHz with a range





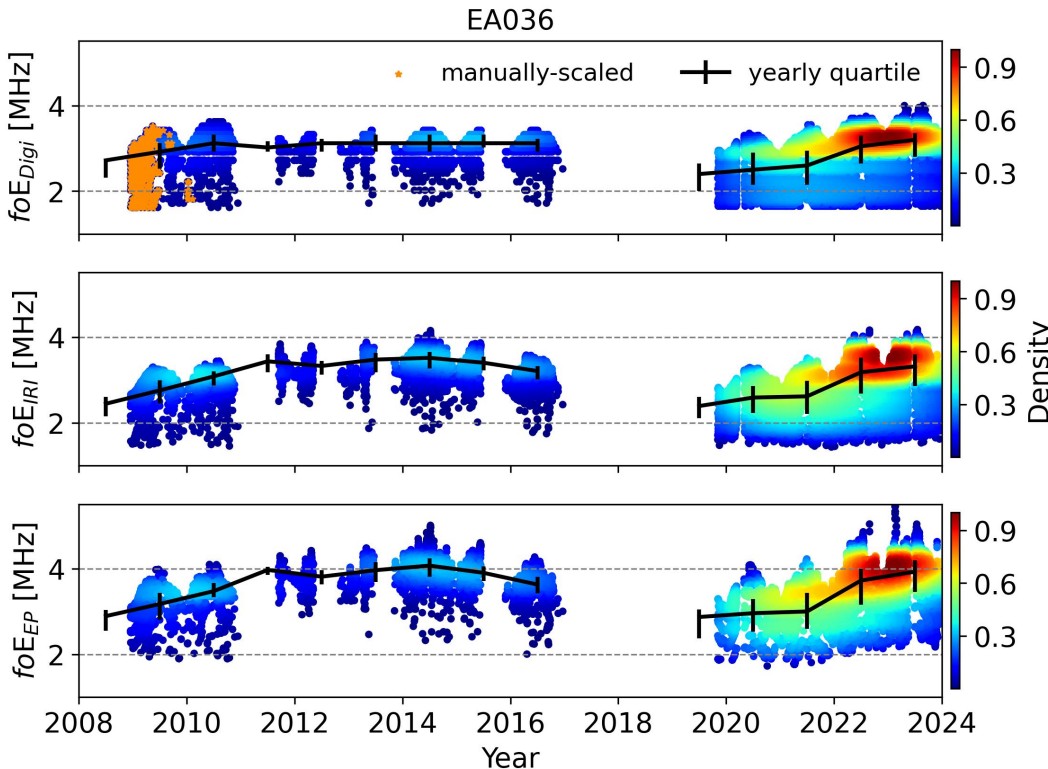

**Figure 1.** Yearly $fo$E estimates for El Arenosillo using ionograms (top), PyIRI (middle), and E-PROBED (bottom). The manually-scaled ionograms are marked with orange stars, and the color represents the normalized data density. Black trend lines intersect the yearly medians, with quartiles (25% and 75%) shown as error bars.

of 2.3–5.0 MHz around 2014. A sharp increase in $fo$E values is observed from 2020 to 2023 for both PyIRI and E-PROBED, which is consistent with the ionosonde trend. In general, $foE_{EP} > foE_{IRI} > foE_{Digi}$, with a larger difference between E-PROBED and ionosondes during Solar Cycle 24 compared to Solar Cycle 25. Both PyIRI and E-PROBED predict multiple spikes in 2023, likely driven by the wide variations in F10.7 ranging from 115–335 sfu during the year. The large $fo$E spike near 5 MHz for E-PROBED in early 2023 corresponds to the observed 335 sfu spike in F10.7. PyIRI's $fo$E calculation goes as F10.7$^{1/4}$ (Equation 15 of Forsythe et al. (2024)), which helps to reduce the impact of this F10.7 spike, better matching the observed ionosonde trends.

Mean Relative Absolute Error (MRAE) calculations were performed for $fo$E following $\mathrm{MRAE} = |fo\mathrm{E}_{model} - fo\mathrm{E}_{obs}|/fo\mathrm{E}_{obs}$. Due to the wide range of errors between models over time and the fact that both models tend to overestimate $fo$E, the absolute error was computed instead of the signed error so that $\log(\mathrm{MRAE})$ could be displayed for comparison. The MRAE was averaged for each month and solar local hour bin, with at least 10 points required for the bin to display a result. As shown in Figure 2, E-PROBED has the largest relative errors near dusk during local summer, with a peak MRAE of 76%. Dawn and





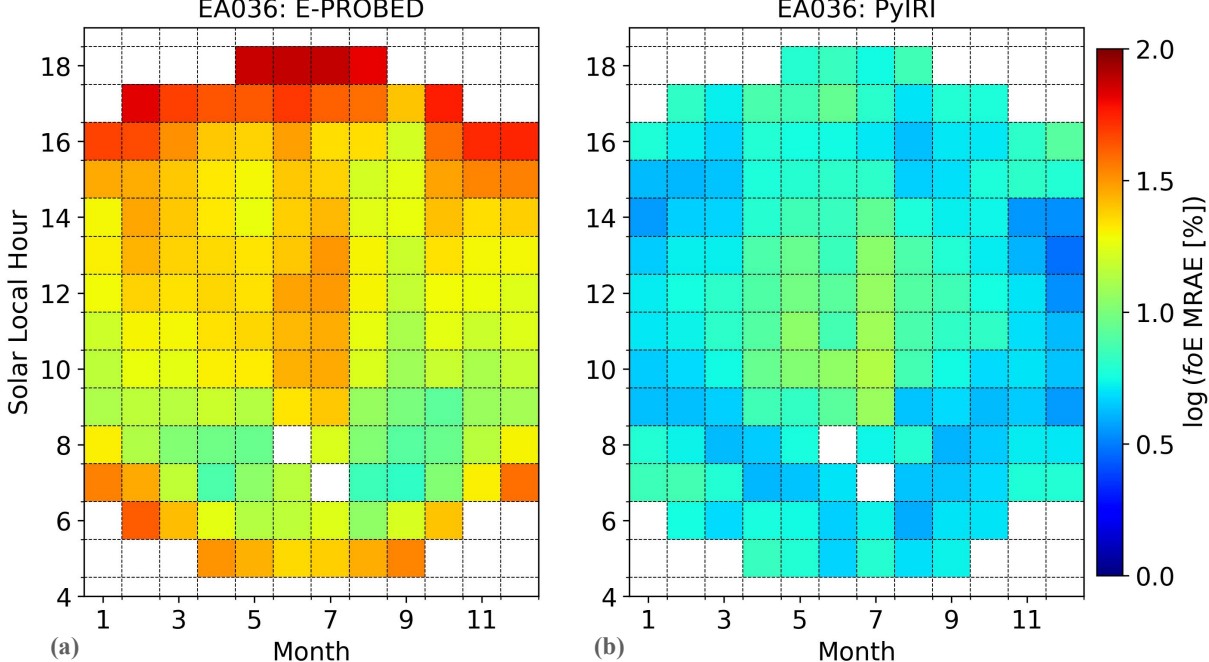

**Figure 2.** Mean Relative Absolute Error (MRAE) for $fo$E predictions by (a) E-PROBED and (b) PyIRI at El Arenosillo. Peak MRAE of 76% is observed near dusk during local summer for E-PROBED while PyIRI MRAE peaks at 13% near noon during summer.

summertime noon also have larger errors for E-PROBED, and the mean MRAE over all times is 24% due to the overestimated foE magnitudes shown in Figure 1. The MRAE for PyIRI are very low in comparison, with peak MRAE of 13% during summertime noon and a time-averaged MRAE of only 6%.

    While $fo$E can be estimated directly from E-region cusps in ionograms, $hm$E requires a best-fit to the entirety of the E-region observations. This difference will be explored in more detail in Section 4, but it is important to point out that the relative uncer-

tainty in $hm$E estimates from ionosondes is generally greater than the $fo$E uncertainties due to the assumption of a parabolic bottomside E-region profile for a Chapman layer and the best-fit procedure required to extract the peak frequency, height, and layer thickness (Reinisch and Xueqin, 1983; Titheridge, 1985a). From this, while the manually-scaled $fo$E estimates were consistent with the auto-scaled estimates, the manually-scaled $hm$E estimates show significant differences with the auto-scaled values. This difference is readily observed in the yearly $hm$E trends shown in Figure 3. In 2009, the observations with a large

collection of manually-scaled ionograms show median $hm$E values near 115 km, with a significant drop to values near 100 km for the remainder of the comparison period for the auto-scaled ionograms. Manually-scaled ionogram $hm$E values range from 95–135 km, while the auto-scaled ionograms range from 90–115 during Solar Cycle 25.

    The solar cycle variation is less pronounced for $hm$E, but seasonal trends show peaks in the local (boreal) winter that match the expected reciprocal relationship expected between $fo$E and $hm$E for a Chapman layer (Chapman, 1931). E-PROBED $hm$E

predictions also show a less pronounced solar cycle variation compared to $fo$E. The median $hm$E predictions from E-PROBED





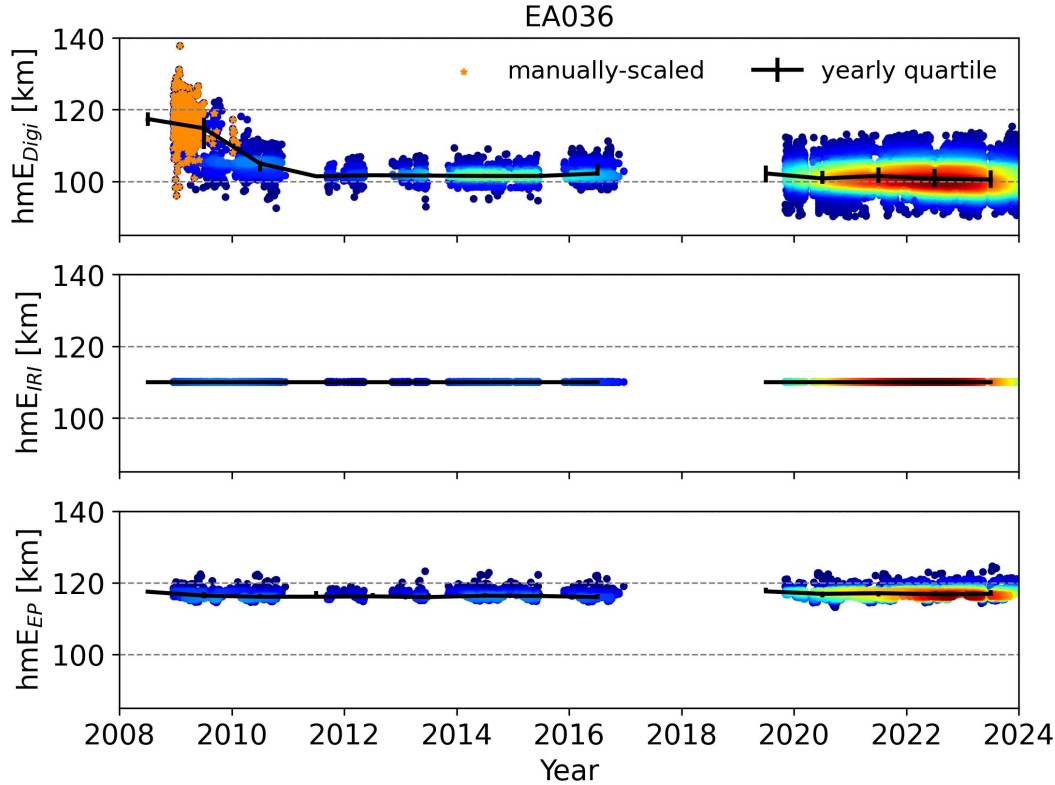

**Figure 3.** Yearly $hm$E estimates for El Arenosillo using ionograms (top), PyIRI (middle), and E-PROBED (bottom). Black trend lines intersect the yearly medians, with quartiles (25% and 75%) shown as error bars.

are consistently around 115 km, which agrees with the manually-scaled ionograms. E-PROBED shows less variation, however, with most predictions between 115–120 km.

PyIRI predicts a constant value of 110 km for all conditions and times in this comparison. For this reason, the PyIRI $hm$E are not displayed in the remaining figures. However, it should be noted that the large number of ionograms removed during quality control with $hm$E $= 110$ km follows from the starting point of 110 km for $hm$E, as suggested by the Committee Consultative for Ionospheric Radiowave Propagation (CCIR) during ionogram inversion with ARTIST (Bradley and Dudeney, 1973; Reinisch et al., 1988). Given the large difference between manually-scaled and auto-scaled ionogram $hm$E estimates along with the constant PyIRI $hm$E value, the remaining $hm$E figures for all sites are reserved for Appendix A.

While $fo$E and $hm$E are helpful parameters to characterize the peak of the E-region, they do not inherently contain information on the shape of the E-region profile. However, virtual height observations provide a method for model comparison that depends on both profile shapes and magnitudes. Since the ionosonde virtual heights are direct observations, this removes the uncertainties created during the ionogram inversion process to provide a more direct comparison with measurements. Virtual heights for an individual layer (e.g., E-layer) are expected to monotonically increase, but the rate of increase depends on the





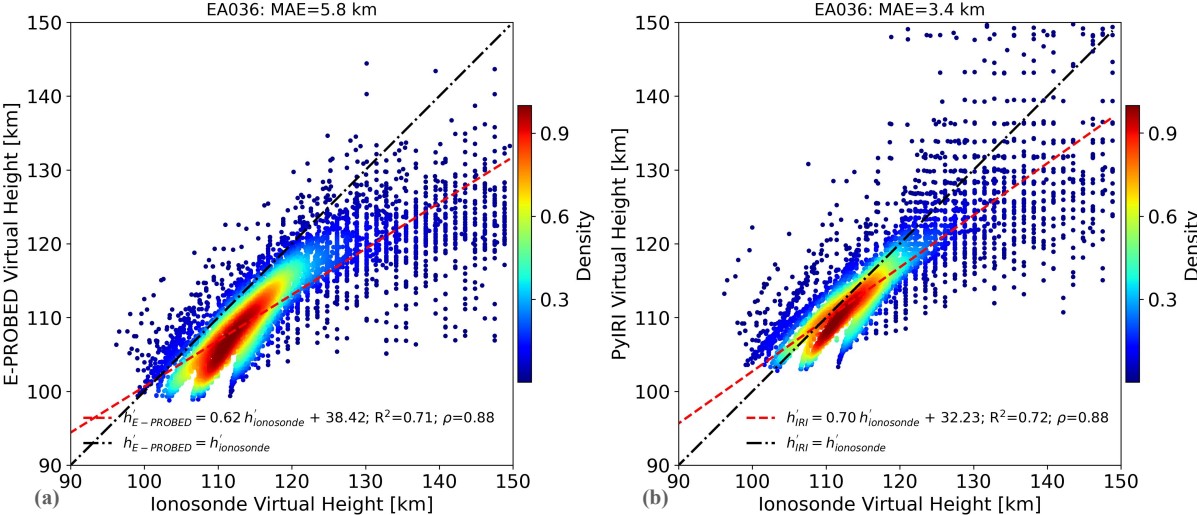

**Figure 4.** Modeled virtual heights from (a) E-PROBED and (b) PyIRI compared against ionosonde observations for El Arenosillo. Observations from 618 (528 manually-scaled) ionograms are displayed spanning Jan–Mar 2009.

spatially integrated group index of refraction, which is a function of the electron density over altitude (Budden, 1966). From
this dependence, differences in electron density magnitudes, altitudes, and shapes will result in virtual height differences such
that calculated virtual heights can be used as a validating metric for modeled EDPs that is free from uncertainties incurred by
profile shape assumptions used during ionogram inversion (Shaver et al., 2023).

The virtual heights derived from E-PROBED and PyIRI over EA036 are shown in Figure 4 compared to ionosonde observations. The period of Jan–Mar 2009 was selected for comparison because it contained a large density of manually-scaled
ionograms (528 of the 618 total). Each point in the figure corresponds to the virtual height measured by the ionosonde and the
modeled virtual height using the transmitted ionosonde frequency for all frequencies below $foE$ in each individual ionogram.
With the roughly 25 kHz transmit frequency resolution, this corresponds to nearly 8000 datapoints for comparison.

Both PyIRI and E-PROBED match the ionosonde observations fairly well, with $R^2$ values of 0.7 and a Spearman's rank
correlation coefficient, $\rho$, of 0.9 for both. The overall agreement is better for PyIRI with a Mean Absolute Error (MAE) of 3 km
and a linear fit slope of 0.7 while E-PROBED produces an MAE of 6 km and a slope of 0.6. Both models show a slight underestimation for the majority of predictions, and the reduced linear fit slope for E-PROBED is due to a collection of underestimated
virtual heights for ionosonde virtual heights above 130 km. This underestimation stems from the slight overestimation of $foE$
values, which pushes the modeled $foE$ cusps to frequencies higher than those observed in the ionosondes. The larger virtual
heights correspond to frequencies approaching the E-region cusp, such that variations in $foE$ can map to relatively large errors
in virtual heights.

Interestingly, the virtual height predictions from PyIRI align very well with the observations, likely due to the close agreement in $foE$ even though a constant $hmE$ value of 110 km is estimated for every profile. As virtual heights are dependent





on the integral of the altitude ($z$) gradient with respect to the plasma frequency ($f_p$), $\mathrm{d}z/\mathrm{d}f_p$, the shape of the profiles plays an important role (Budden, 1966). From this, an underestimation bias in $hm$E by PyIRI (see manually-scaled ionogram data

in Figure 3) can be compensated for with elevated $\mathrm{d}z/\mathrm{d}f_p$ for a given transmit frequency to produce similar virtual heights. This $\mathrm{d}z/\mathrm{d}f_p$ is a function of $fo$E and layer semi-thickness, which are, in fact, adjusted during ionogram inversion to match observed virtual heights (Reinisch et al., 1988).

### 3.2 Fortaleza, Brazil

To avoid duplicated discussion, we focus on comparing and contrasting trends with EA036 instead of focusing on specific
values in the discussion below. The yearly $fo$E trends for FZA0M follow the same trends as observed over EA036 with a clear solar cycle variation showing elevated $fo$E values during solar maximum (Figure 5). E-PROBED $fo$E estimates are greater than PyIRI and ionosonde values, while the latter two are nearly equal on average. A spike is observed in September 2017 for both E-PROBED and PyIRI when F10.7 increased to 185 sfu. However, this $fo$E spike is not observed in the ionograms. It must be noted that this equatorial region is prone to E-region ionospheric irregularities from equatorial electrojet instabilities
(Arras et al., 2022) and particle precipitation allowed by the South Atlantic Anomaly (Moro et al., 2022), which can cause additional uncertainties in ionogram auto-scaling.

MRAE calculations for modeled $fo$E are shown in Figure 6. Similar to EA036, E-PROBED shows larger relative errors with a peak MRAE of 64% at dusk and a time-averaged MRAE of 19%. The PyIRI errors are much lower, peaking at 7% during the afternoon with a time-averaged MRAE of 5%. Seasonal MRAE variations are less pronounced, as expected for this equatorial
site.

Due to ambiguities between manually-scaled and auto-scaled $hm$E, the yearly $hm$E trends are reserved for Appendix A. For the virtual height comparison, a total of 711 ionograms from Aug 2019 were used as the ground-truth (Figure 7). While the mid-latitude EA036 site showed relatively strong agreement between the modeled and measured virtual heights for both E-PROBED and PyIRI (Figure 4), the equatorial FZA0M virtual height agreement is weaker. A linear fit of the E-PROBED
virtual heights produces a slope of approximately 0.5 with an $R^2$ of 0.6, a Spearman's rank correlation coefficient of 0.8, and an MAE of 5 km. PyIRI produces similar $R^2$ and Spearman's rank correlation coefficient values, but with an MAE of 7 km and a linear fit slope of 0.6.

Both E-PROBED and PyIRI show a positive virtual height bias, unlike the negative bias produced for EA036. This change is related to the difference in manually-scaled vs auto-scaled ionogram $hm$E estimates, where the manually-scaled $hm$E for
EA036 are ~15 km above the corresponding auto-scaled estimates. This reduction in ionosonde $hm$E corresponds to reduced virtual heights, thereby creating a positive bias in the modeled virtual heights for FZA0M.

### 3.3 Gakona, United States

Similar to the previous subsection, here we focus on general trends and differences between GA762 results and the results of EA036 and FZA0M. As observed in Figure A8, the $fo$E estimates for both PyIRI and E-PROBED show solar cycle trends
similar to those of the ionosonde observations. However, from 2022-2024, E-PROBED predicts a steady increase in $fo$E



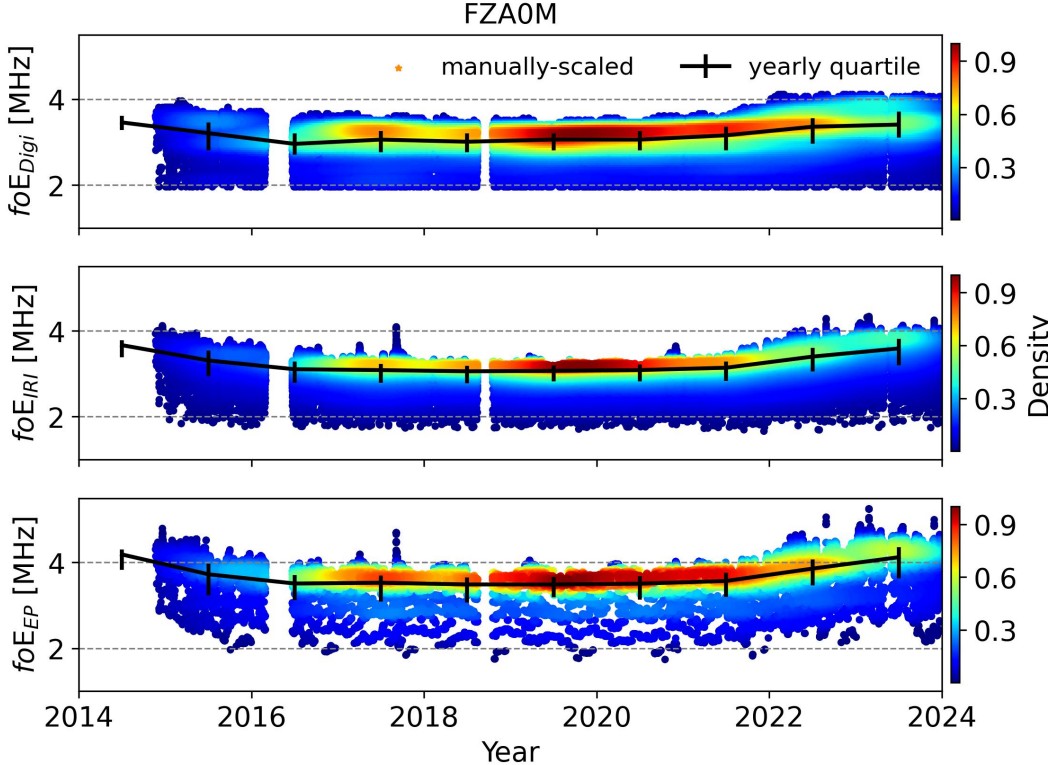

**Figure 5.** Yearly $foE$ estimates for Fortaleza using ionograms (top), PyIRI (middle), and E-PROBED (bottom). Black trend lines intersect the yearly medians, with quartiles (25% and 75%) shown as error bars.

while the ionosonde and PyIRI trends remain flat over time. As for the previous sites, E-PROBED slightly overpredicts $foE$ while PyIRI is nearly equal (with a very slight positive bias). It should be noted that the dynamic ionization contribution from precipitating electrons (Solomon, 1993) is difficult to capture in climatological models (Themens and Jayachandran, 2016), such that periods with elevated electron flux may reduce the gap between model overpredictions and ionosonde $foE$
observations.

MRAE for the $foE$ predictions are shown in Figure 9. For Gakona, E-PROBED shows the largest MRAE of 58% near autumnal dusk while also showing large errors near dusk throughout the year. PyIRI produces peak MRAE of 12% near summertime dusk, with a low time-averaged MRAE of 7% compared to 19% for E-PROBED.

Modeled virtual heights for GA762 show a slightly larger slope of 0.6 for the E-PROBED predictions compared to 0.5 for
PyIRI (Figure 10). However, the E-PROBED predictions also show more variance with lower $R^2$ and $\rho$, mostly caused by the spread in the predictions and measurements for larger virtual heights above 130 km. This wider variance maps to a slightly larger MAE of 7 km for E-PROBED compared to the 6 km for PyIRI.



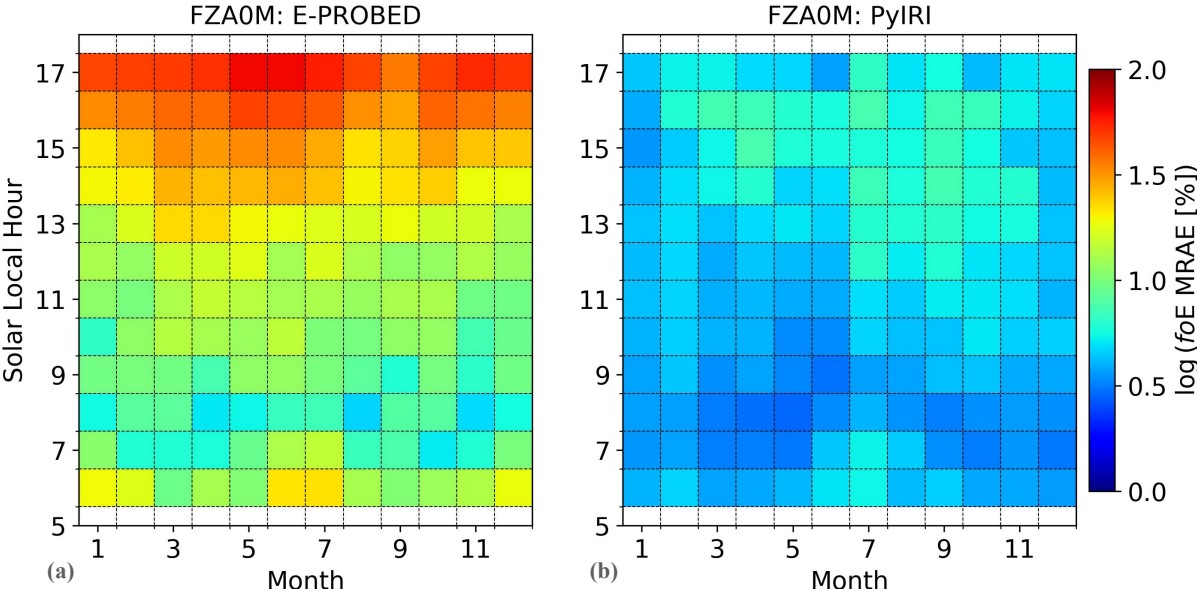

**Figure 6.** Mean Relative Absolute Error (MRAE) for $foE$ predictions by (a) E-PROBED and (b) PyIRI at Fortaleza. Peak MRAE of 64% is observed near dusk for E-PROBED while PyIRI MRAE peaks at 7% in the afternoon.

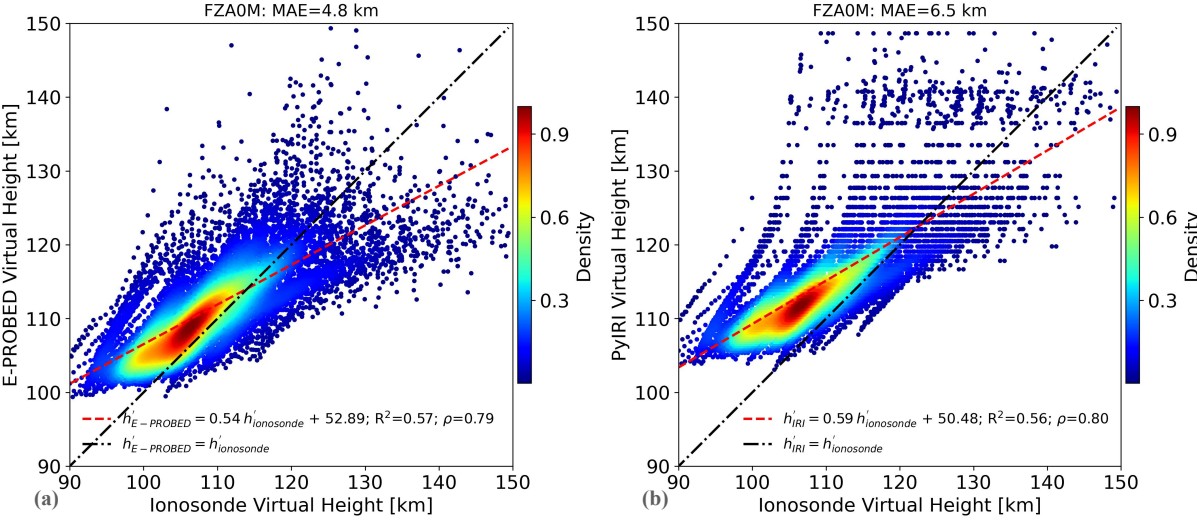

**Figure 7.** Modeled virtual heights from (a) E-PROBED and (b) PyIRI compared against ionosonde observations for Fortaleza. Observations from 711 ionograms are displayed spanning Aug 2019.

Both E-PROBED and PyIRI generally overestimate the virtual heights, similar to the overestimated $hmE$ altitudes (Figure A10). While PyIRI holds a constant $hmE$ of 110 km, the close match in $foE$ results in relatively close agreement for predicted





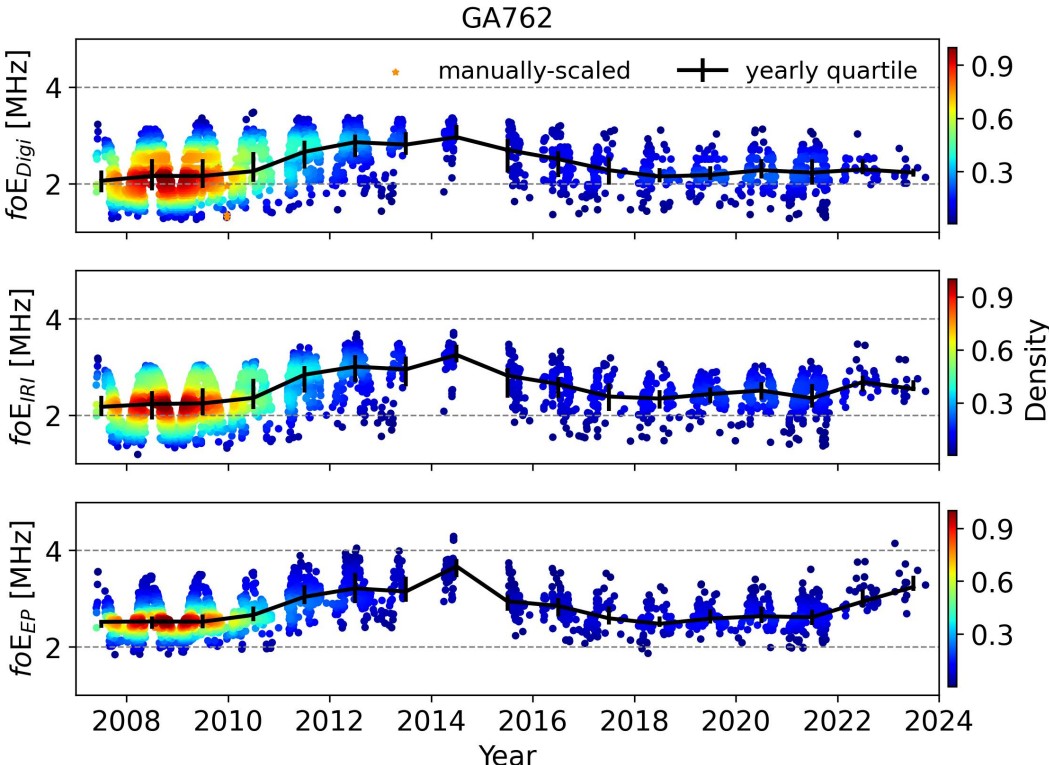

**Figure 8.** Yearly $fo$E estimates for Gakona using ionograms (top), PyIRI (middle), and E-PROBED (bottom). Black trend lines intersect the yearly medians, with quartiles (25% and 75%) shown as error bars.

virtual heights. The larger variance in E-PROBED virtual heights is due to the larger spread in predicted $fo$E and $hm$E values over Gakona.

## 4   Discussion

Overall, both E-PROBED and PyIRI show reasonable agreement with the ionosonde observations spanning low-, mid-, and high-latitudes; the combined statistics are displayed in Table 1. Between $fo$E and $hm$E, both models show better agreement

with $fo$E. PyIRI shows close $fo$E agreement with ionosonde observations producing MAE values between 0.1–0.2 MHz. E-PROBED's $fo$E MAE values are slightly larger (0.4–0.5 MHz), but still within a reasonable range of the ionosonde observations. For comparison, an $fo$E uncertainty of $\pm 0.3$ MHz is estimated by (Reinisch and Xueqin, 1983) for ionogram inversion.

PyIRI calculates $fo$E in a manner similar to NeQuick (Nava et al., 2008), somewhat different from IRI, with the magnitude of $fo$E as a function of the effective SZA, a seasonal parameter, and the F10.7 solar radio flux (Forsythe et al., 2024).

This NeQuick $fo$E relationship was adopted from Titheridge (1996), which used a photochemistry-based model in compar-



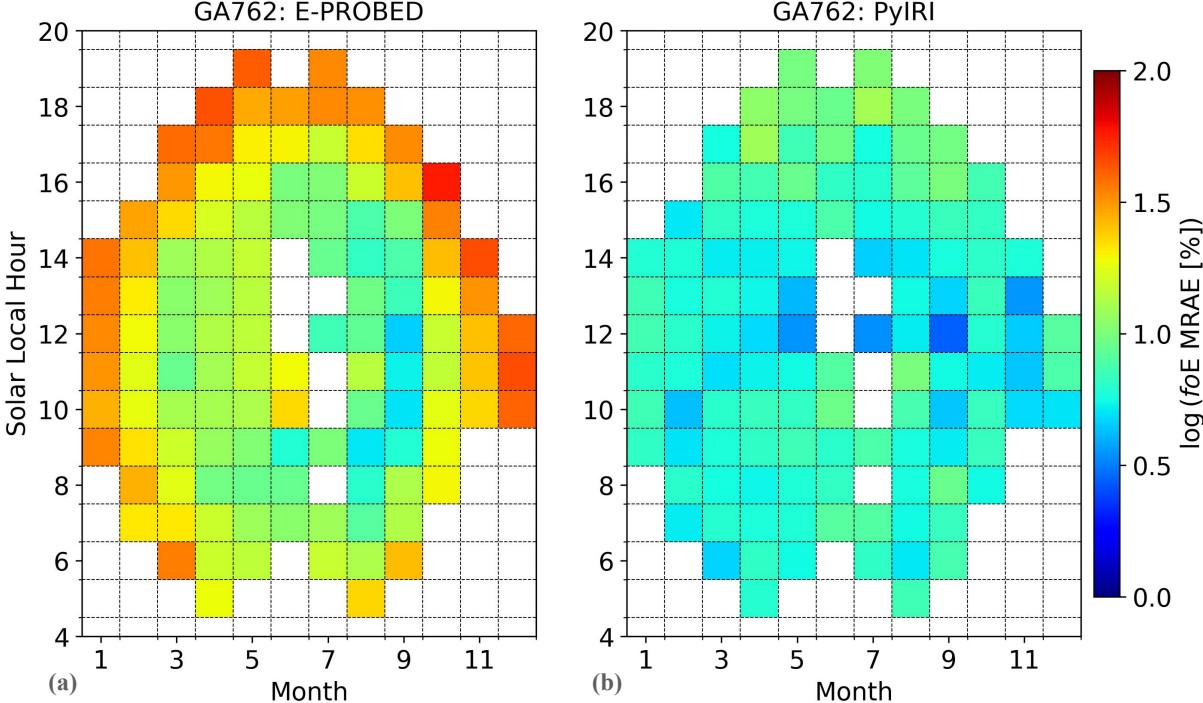

**Figure 9.** Mean Relative Absolute Error (MRAE) for $foE$ predictions by (a) E-PROBED and (b) PyIRI at Gakona. Peak MRAE of 58% is observed near dusk during autumn for E-PROBED while PyIRI MRAE peaks at 12% near dusk in the summer.

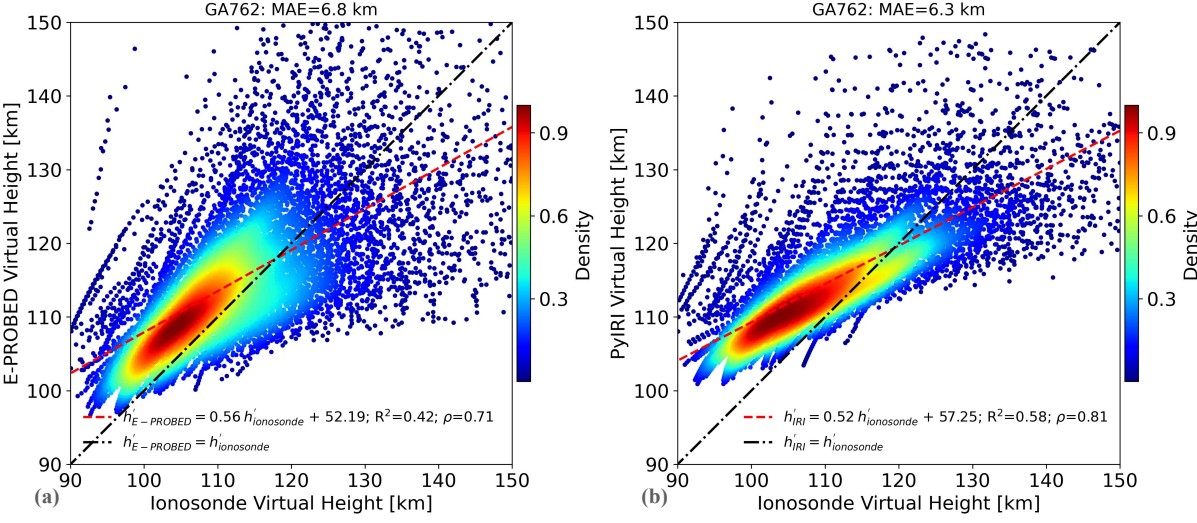

**Figure 10.** Modeled virtual heights from (a) E-PROBED and (b) PyIRI compared against ionosonde observations for Gakona. Observations from 515 ionograms are displayed spanning May 2008 to Jan 2009.





ison against ionosonde-based IRI and results from Kouris and Muggleton (1973b). In contrast, E-PROBED is derived from COSMIC-1 radio occultation observations and is driven by SZA, season, and F10.7, with an additional non-SZA component that is a function of latitude, local time, season, and F10.7 (Salinas et al., 2024). Although the $foE$ ($NmE$) Fourier coefficient calculations from E-PROBED are more complicated than the analytical PyIRI approach, they also provide more variability

over time and latitude (see Figure 9 of Salinas et al. (2024)).

It should be noted that the largest MRAE for E-PROBED $foE$ predictions occurred near dusk when large ionospheric tilts are present that impact the HF ray-paths used to create ionograms (McNamara, 1991). These ionospheric tilts result in off-zenith observations, inducing additional errors into the ionogram inversion process that assumes vertical propagation (Reinisch and Xueqin, 1982). The elevated MRAE for Gakona in winter may also be influenced by this phenomenon, where the short period

of sunlight can be considered dawn/dusk instead of daytime. PyIRI's close relationship with historical ionosonde data likely incorporates the impact of the ionospheric tilts on the dawn/dusk ionograms, as observed for the low dawn/dusk MRAE shown here. However, as outlined by Paznukhov et al. (2020), the largest zenith variations are observed during sunrise, whereas the variation during sunset is minimal, suggesting that ionospheric tilt-induced uncertainties in ionogram observations are probably not the cause of larger E-PROBED errors near dusk.

For $hmE$, the PyIRI constant $hmE$ estimate of 110 km results in lower MAE values than the variable $hmE$ predictions from E-PROBED. However, the resulting $R^2$ and Spearman rank-order correlation coefficients ($\rho$) for PyIRI's $hmE$ predictions are essentially zero, while $\rho$ for E-PROBED $hmE$ ranges from 0.11-0.25. The constant daytime $hmE$ of 110 km in PyIRI originates from IRI's constant value of 105 km that was changed to "110 km based on input from ionosonde and ISR observations" (Bilitza et al., 2022). As suggested by Titheridge (2000), these "reflect the uncertainty caused by a lack of good observational data,"

where good ionograms before the year 2000 could only be scaled with a virtual height accuracy of 2–3 km. Even the modern Digisonde 4D has a range resolution of 2.5 km (Galkin et al., 2009). However, as developed by Ivanov-Kholodny et al. (1998) from ISR observations and Titheridge (2000) from ionosondes, time-varying $hmE$ models are currently available showing $hmE$ variations between 105-120 km, which may be beneficial for implementation in IRI/PyIRI.

The large difference between auto-scaled and manually-scaled $hmE$ estimates as observed in Figure A3 must be taken

into account to fully understand the uncertainties involved with auto-scaled ionogram parameters. The manually-scaled $hmE$ estimates over EA036 are nearly 15 km above the auto-scaled values for the same conditions, closely matching the E-PROBED $hmE$ estimates. ARTIST-5 auto-scaling uncertainties have been analyzed in detail by Stankov et al. (2023) for the Dourbes, Belgium Digisonde during 2011-2017, resulting in $foE$ error bounds (auto-scaled value minus manually-scaled value) of [-0.30,0.80] MHz and minimum virtual height of the E-region, $h'E$, error bounds of [-6,6] km. While the modeled $foE$ values

fall within the auto-scaled ionogram uncertainties for $foE$, the auto-scaled $h'E$ uncertainties are rather low compared to the models' $hmE$ MAE between 6–15 km. However, the auto-scaled $h'E$ error bounds during low solar activity show a large underestimation bias for the auto-scaled estimates ranging from [-15.0,2.5] km (Stankov et al., 2023). The -15 km error bound matches the 15 km $hmE$ underestimation observed during the 2009 solar minimum over EA036, indicating that the large errors in E-PROBED $hmE$ estimates during solar minimum may simply be an artifact of auto-scaled errors. As the auto-scaled

error bounds are reduced during solar maximum, the E-PROBED $hmE$ estimates are overestimated during these times. The





**Table 1.** Statistics for the entire datasets comparing E-PROBED and PyIRI with ionosonde observations. In this comparison, MAE is Mean Absolute Error, $\rho$ is the Spearman rank-order correlation coefficient, and $h'$ is the virtual height. Due to the large differences between manually-scaled and auto-scaled ionogram $hm$E estimates, care must be taken in interpreting the $hm$E results below.

| Model | Site | Parameter | MAE | $R^2$ | $\rho$ |
|---|---|---|---|---|---|
| **E-PROBED** | EA036 | $fo$E | 0.6 MHz | 0.71 | 0.84 |
| | | $hm$E | 14.8 km | 0.03 | 0.25 |
| | | $h'$ | 5.8 km | 0.71 | 0.88 |
| | FZA0M | $fo$E | 0.5 MHz | 0.53 | 0.70 |
| | | $hm$E | 11.3 km | 0.04 | 0.23 |
| | | $h'$ | 4.8 km | 0.57 | 0.79 |
| | GA762 | $fo$E | 0.4 MHz | 0.69 | 0.84 |
| | | $hm$E | 14.9 km | 0.00 | 0.11 |
| | | $h'$ | 6.8 km | 0.42 | 0.71 |
| **PyIRI** | EA036 | $fo$E | 0.2 MHz | 0.87 | 0.91 |
| | | $hm$E | 8.4 km | 0.00 | 0.02 |
| | | $h'$ | 3.4 km | 0.72 | 0.88 |
| | FZA0M | $fo$E | 0.1 MHz | 0.81 | 0.87 |
| | | $hm$E | 5.9 km | 0.00 | 0.00 |
| | | $h'$ | 6.5 km | 0.56 | 0.80 |
| | GA762 | $fo$E | 0.1 MHz | 0.89 | 0.95 |
| | | $hm$E | 8.8 km | 0.00 | 0.03 |
| | | $h'$ | 6.3 km | 0.58 | 0.81 |

general trend of underestimated peak height altitudes from ARTIST-5 auto-scaled ionograms was also observed for $hm$F2 when compared against GNSS-RO and ISR estimates (Swarnalingam et al., 2023), and the $hm$E differences between auto- and manual-scaled ionograms observed in Figures A3–A4 suggest that perhaps an offset could be calculated to correct the auto-scaled estimates. However, this offset would likely depend on several factors such as solar cycle and ionosonde site (hardware,
climatology, etc.), requiring a significant effort to obtain appropriate correction factors.

    It must also be noted that $hm$E is dependent on a parabolic fit to the E-region virtual height observations (Reinisch and Xueqin, 1983; Titheridge, 1985a), which means that the scaling errors for $h'$E and errors in $hm$E are not expected to be one-to-one. However, differences in the starting altitude of virtual height observations will certainly impact the $hm$E estimates, and the differences between manually-scaled and auto-scaled $hm$E observations align well with the expected errors in $h'$E. An
additional uncertainty/error within ionogram-derived $hm$E estimates is caused by the assumed parabolic layer shape for the E-region that approaches a zero plasma frequency at the bottom of the E-region rather than smoothly transition to the nonzero D-region densities. As discussed in detail in Shaver et al. (2023), the lack of a D- to E-region transition during ARTIST-5



ionogram inversion along with a required parabolic profile fit (that can introduce exaggerated $\mathrm{d}z/\mathrm{d}f_p$ near $fo$E) may result in a small bias for the bottom of the parabolic layer on the order of a few kilometers.

Compared with previous comparisons of E-region models with ionosonde observations, the results generally align with the present study. Mikhailov et al. (1999) found similar seasonal trends with the El Arenosillo Digisonde throughout 1995. They also found relatively close agreement with IRI's $fo$E estimates and ionosonde observations, but noted that a chemistry-based model could reproduce the $hm$E variations not captured by IRI. The $fo$E and $hm$E model developed by Titheridge (2000) was able to reduce $hm$E errors to less than 5 km and $fo$E errors below 0.1 MHz when compared to manually-scaled ionograms from

Auckland, New Zealand, which is a significant reduction from the $hm$E errors observed here for E-PROBED and PyIRI. Yue et al. (2006) found that IRI overestimated $fo$E over Wuhan, China, especially between May and September. Pavlov and Pavlova (2013) developed a photochemistry-based $Nm$E model that showed reasonable agreement with an ionosonde at Boulder, Colorado for low solar activity, but required a factor of 2 increase in the 3.2-7.0 nm flux from EUVAC to match observations during high solar activity. A comparison of IRI $fo$E predictions with an ionosonde at Chumphon Station, Thailand, found the

largest differences during sunrise and sunset, but very low overall errors (Wongcharoen et al., 2015). Mostafa et al. (2018) also found a slight overprediction by IRI compared to manually-scaled ionograms from the Nicosia, Cyprus Digisonde. For $hm$E, they observed similar diurnal and seasonal trends, although their manually-scaled ionograms provided $hm$E values below IRI's constant value of 110 km. Further, they conclude that large differences between IRI and the ionosonde observations may be due to non-Chapman like behavior of the E-layer, which was also noted by Ivanov-Kholodny et al. (1998). These results are

consistent with the auto-scaled ionograms shown here, although our manually-scaled ionograms from EA036 show elevated $hm$E values above 110 km.

In contrast to $fo$E and $hm$E estimates from ionogram inversions, virtual heights are directly measured by ionosondes, making them a useful tool for model validation. The virtual height ($h'$) for a particular transmit frequency ($f$) is defined as the integral of the group index of refraction as a function of altitude, which may also be represented as the group index ($\mu'$) as a

function of frequency times the gradient of the real-height with respect to frequency:

$$h'(f) = \int_0^f \mu'(f, f_p) \frac{\mathrm{d}z}{\mathrm{d}f_p} \, \mathrm{d}f_p, \tag{1}$$

$$\mu' = \left(1 - \frac{f_p^2}{f^2}\right)^{-1/2}, \tag{2}$$

where $f_p$ is the plasma frequency (Budden, 1966). Equation 2 is a simplified form of the group index of refraction for an unmagnetized, collisionless plasma, and is shown here instead of the full form for an O-mode to show the basic relationship

with respect to the plasma frequency. This dependence on $f_p$ and $\mathrm{d}z/\mathrm{d}f_p$ provides a method for analyzing modeled E-region shapes and gradients that is free from uncertainties produced by profile shape assumptions used during ionogram inversion. However, due to the integrated nature of the virtual heights, it is possible to produce the same virtual heights from various EDPs, meaning that virtual height agreement does not necessarily indicate agreement on profile shapes overall. Although a direct comparison with ionosonde-derived EDPs may seem like a more straightforward approach, the various assumptions



required for ionogram inversion result in a final product that is no longer a direct measurement (see the discussion in Shaver et al. (2023)).

For the sites analyzed here, both PyIRI and E-PROBED showed reasonable agreement with the measured virtual heights with MAE ranging from 5–7 km for E-PROBED and 3–7 km for PyIRI (Table 1). Although similar performance for both models may appear to indicate similar profiles that agree with ionosonde observations, the differences in predicted $foE$ and
$hmE$ prove that this is not the case. Even with differing EDPs, the altitude integrals of the group indices derived from the EDPs result in similar virtual heights that generally agree with ionosonde observations. Although, a bias exists for each site: the models tend to underestimate the virtual heights for EA036, while the models tend to overestimate the virtual heights for FZA0M and GA762. Since the EA036 period of comparison for virtual heights was Jan–Mar 2009 (solar minimum) and was mostly composed of manually-scaled ionograms while FZA0M and GA762 consisted of auto-scaled ionograms, the change in
bias from underestimation to overestimation is likely an artifact of ARTIST-5 autoscaled uncertainties for $h'E$ (Stankov et al., 2023). Even here, where we assume that the virtual heights are direct measurements to be used as ground-truth, an uncertainty exists from auto-scaling that must be considered when interpreting the accuracy of the modeled virtual heights.

## 5   Conclusions

A comparison of E-region predictions from E-PROBED, PyIRI and ionosondes was performed for three sites: mid-latitude El
Arenosillo, Spain (EA036), low-latitude Fortaleza, Brazil (FZA0M), and high-latitude Gakona, Alaska (GA762). Manually-scaled or auto-scaled ionograms using ARTIST-5 were used as the ground-truth for $foE$ and $hmE$ estimates, and both models were run for each ionogram time spanning from 2009-2024 for EA036 and GA762, and 2015-2024 for FZA0M. Additionally, a subset of ionograms were used to compare against modeled virtual heights calculated from the models using a numerical ray tracer.

The key results of the comparison are listed below:

- Overall, both E-PROBED and PyIRI showed reasonable agreement with the ionosonde observations, properly capturing the $foE$ solar cycle, seasonal, and diurnal trends.

- For $foE$, the E-PROBED estimates were generally larger than the PyIRI estimates, which were slightly larger but nearly equal to the ionosonde observations. Mean Relative Absolute Errors (MRAEs) peaked around 70% for E-PROBED at
dusk, while PyIRI produced lower MRAE peaks around 10% for times ranging from late morning to dusk.

- For $hmE$, both models showed weaker agreement with auto-scaled ionograms; E-PROBED overestimated by ∼15 km and PyIRI predicted a constant $hmE$ of 110 km. The large bias in E-PROBED $hmE$ estimates almost disappears when compared to manually-scaled ionograms, indicating that great care must be taken when comparing against auto-scaled $hmE$ estimates.

- Modeled virtual heights derived from E-PROBED and PyIRI showed reasonable agreement with measured virtual heights overall. Since ionosondes measure virtual heights directly, this comparison provides confidence in the integrated





electron density profiles as a function of altitude. A slight bias exists in the modeled virtual heights that reverses direction for manual- versus auto-scaled ionograms, indicating that auto-scaled uncertainties are also present in the virtual height observations, similar to $hm$E.

This comparison provides confidence in the use of E-PROBED and PyIRI for global E-region predictions. While both models can be improved in future iterations, the solar cycle, seasonal, and diurnal trends were captured well overall. A similar study using only manually-scaled ionograms for E-region observations would be helpful, especially given the large ambiguities arising from differences in the manually-scaled and auto-scaled ionogram estimates of $hm$E.

*Code and data availability.*    The ionosonde data used here can be obtained from the Digital Ionogram Database (DIDBASE): https://giro.
uml.edu/didbase/. E-PROBED version 1.0 can be obtained from https://github.com/ccjsalinasNASA/EPROBED_v01.00, and PyIRI can be downloaded from https://pyiri.readthedocs.io/en/latest/overview.html.

*Author contributions.*    DJE, CCJHS, DLW, NS, EVD, JLC, YY, and KEF developed the methodology. DJE analyzed the data with input and feedback from CCJHS, DLW, NS, EVD, JLC, YY, and KEF; DJE and wrote the manuscript draft; CCJHS, DLW, NS, EVD, JLC, YY, and KEF reviewed and edited the manuscript.

*Competing interests.*    No competing interests were present in the research or writing of this manuscript.

*Acknowledgements.*    This research was funded by NASA's Living With a Star (LWS) and Commercial Smallsat Data Acquisition (CSDA) programs, WBS numbers 936723.02.01.12.48 and 880292.04.02.01.68. We thank the Global Ionosphere Radio Observatory (GIRO) for the use of their data.



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

## Appendix A: Seasonal and Diurnal Results

## A1 EA036

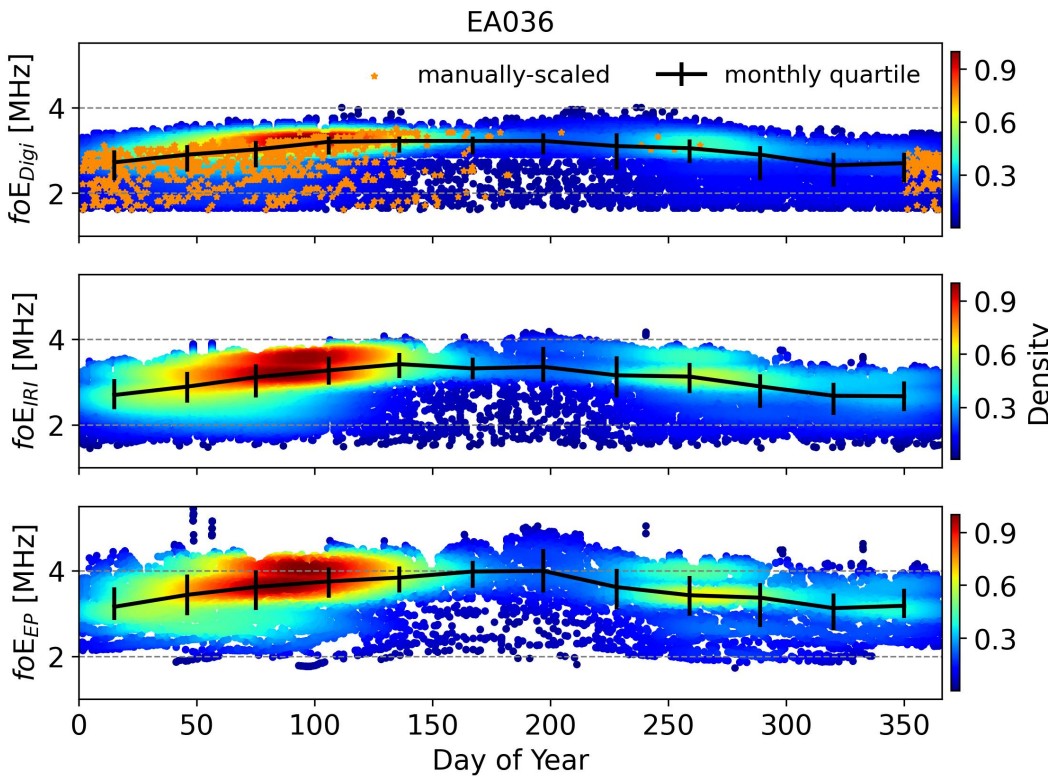

**Figure A1.** Day of Year $fo$E estimates for El Arenosillo using ionograms (top), PyIRI (middle), and E-PROBED (bottom). Black trend lines intersect the monthly medians, with quartiles (25% and 75%) shown as error bars.

Seasonal $fo$E trends are displayed in Figure A1 with monthly quartiles. While the manually-scaled $fo$E values show a lower maximum than the auto-scaled results, it must be noted that a variety of solar cycle conditions are shown here and the auto-scaled ionograms are constrained to 2009. A seasonal variation is readily observed with $fo$E peaks in local summer and troughs in local winter. Median ionosonde observations range from 2.7 MHz in the winter to 3.2 MHz in the summer. Similarly, 520 PyIRI median $fo$E values range from 2.7–3.4 MHz, and E-PROBED ranges from 3.1–4.0 MHz. The same seasonal trends are observed between the models and ionosonde observations, with a slight $fo$E overestimation from E-PROBED.





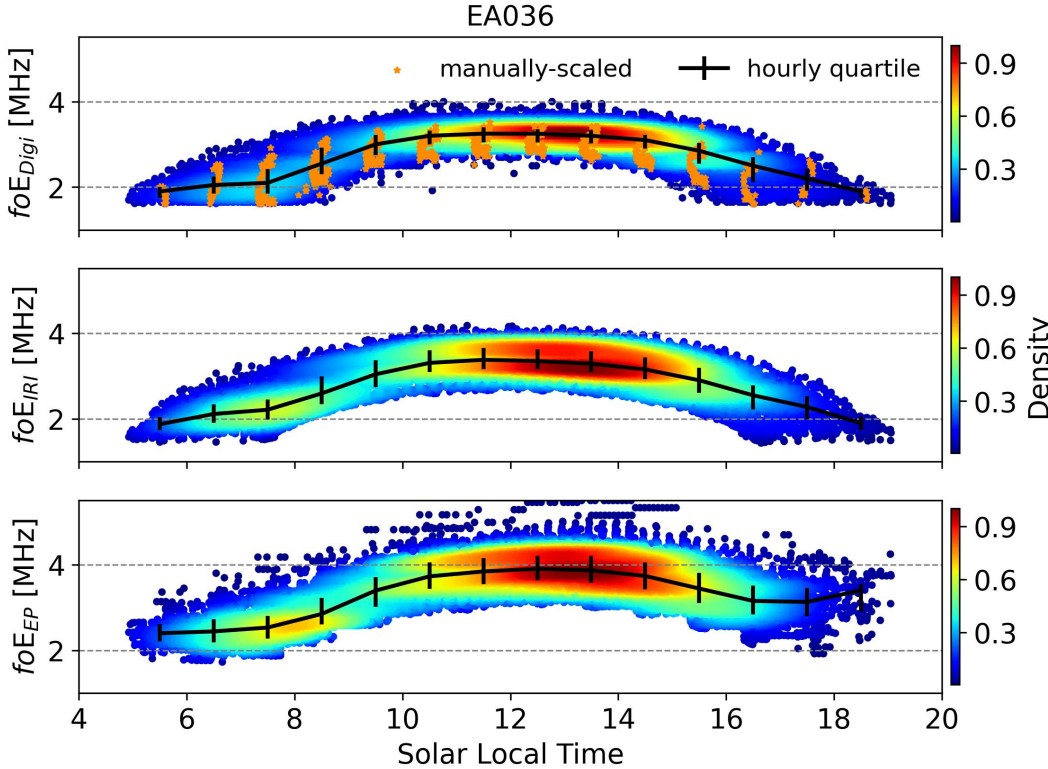

**Figure A2.** Solar local time $foE$ estimates for El Arenosillo using ionograms (top), PyIRI (middle), and E-PROBED (bottom). Black trend lines intersect the hourly medians, with quartiles (25% and 75%) shown as error bars.

Diurnal $foE$ trends are shown in Figure A2. As expected, the $foE$ peaks in the early afternoon following the peak in solar Extreme Ultraviolet (EUV) flux, with minima observed near dawn/dusk. Median ionosonde $foE$ measurements range from 1.9 MHz at dawn/dusk to 3.2 MHz in the early afternoon. A slow increase is observed until 0800 solar local, when the $foE$
increases more rapidly towards the peak. This general trend is also predicted by PyIRI and E-PROBED. PyIRI medians range from 1.9–3.4 MHz, while E-PROBED slightly overestimates with a range of 2.4–3.9 MHz. Interestingly, E-PROBED show an evening minima around 1700 solar local, followed by a slow $foE$ increase later in the evening.

The seasonal $hmE$ trends from ionosondes show a slight decrease during local summer when the $foE$ values peak (Figure A3), as expected for a Chapman layer. However, this decrease is relatively small, with median values ranging from 103 km in
winter to 100 km in summer. E-PROBED also shows slight seasonal variation with $hmE$ peaks during winter and summer. The medians range from 115–117 km, with a larger spread in predictions during the local summer. E-PROBED $hmE$ predictions generally align with the manually-scaled ionograms, which are nearly 15 km above the auto-scaled values.



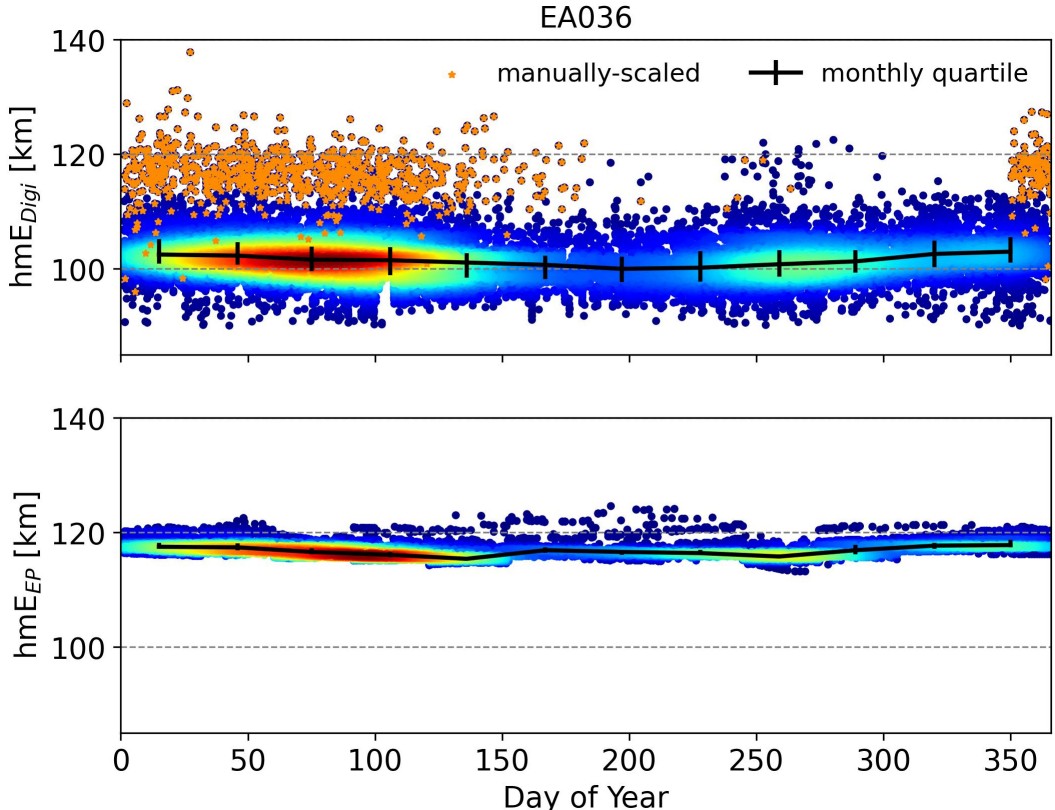

**Figure A3.** Day of Year $hmE$ estimates for El Arenosillo using ionograms (top) and E-PROBED (bottom). Black trend lines intersect the monthly medians, with quartiles (25% and 75%) shown as error bars.

Solar local time $hmE$ variations (Figure A4) show minima near local noon and slight increases toward dusk/dawn. Similar to the seasonal trends, the diurnal changes are relatively small (2-3 km) for both ionosondes and E-PROBED with general agreement on the altitudes for the manually-scaled ionograms.

## A2   FZA0M

The seasonal trends are less pronounced for this equatorial site compared to the mid-latitude EA036, and both models agree with the relatively small seasonal variations from ionosonde observations (Figure A5). E-PROBED shows the largest range of predictions throughout day of year, although the variation does not appear to follow a seasonal trend. The ionosonde and PyIRI $foE$ estimates are generally constant throughout the year.

For the diurnal variation, the ionosonde and PyIRI estimates show a strong symmetry around local noon (Figure A6). In contrast, E-PROBED predicts a dawn-dusk asymmetry with a dawn median of 2.4 MHz and a dusk median of 3.1 MHz.





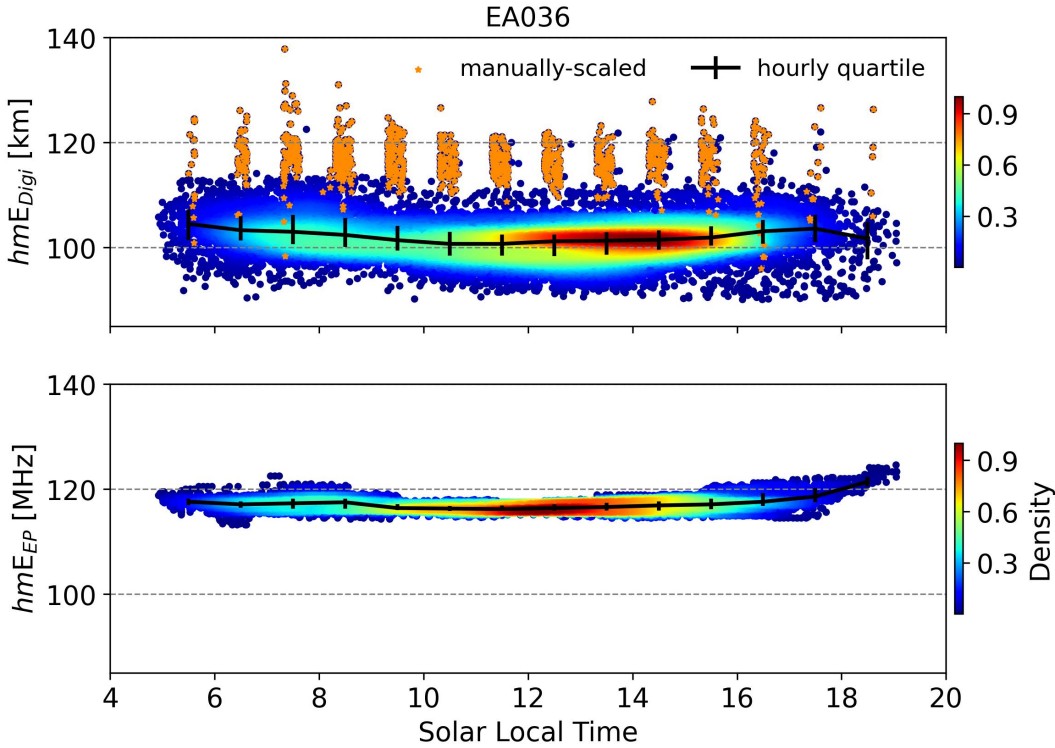

**Figure A4.** Solar local time $hm$E estimates for El Arenosillo using ionograms (top) and E-PROBED (bottom). Black trend lines intersect the hourly medians, with quartiles (25% and 75%) shown as error bars.

Solar cycle $hm$E trends for Fortaleza are displayed in Figure A7, showing the expected increase in altitude during solar minimum when the $fo$E values are reduced. Similar to the EA036 auto-scaled ionogram observations, median $hm$E altitudes are below ~105 km with a range of 90–120 km, while PyIRI predicts a constant $hm$E of 110 km and E-PROBED predicts median values around 115 km with a reduced range.

## A3    GA762

Seasonal trends are similar between both models and the ionosonde observations (Figure A8), with a slight flattening toward local winter observed in the E-PROBED trends. The diurnal variation is less pronounced at this high-latitude site, but $fo$E peaks are still observed near local noon (Figure A9). Of note, both PyIRI and the ionosonde estimates show a double peak in data density near local noon (two distinct $fo$E peaks), while E-PROBED has a single peak. This double peak in data density may be due to seasonal variations, where PyIRI and ionosondes have a higher data density at elevated $fo$E near the local summer, while the summer E-PROBED estimates are relatively low compared to the winter values (Figure A8).

The $hm$E trends for GA762 are similar to EA036 and FZA0M with E-PROBED estimates larger than PyIRI (which are held at a constant 110 km), which is greater than the auto-scaled ionogram estimates (Figure A10). Subtle solar cycle variations



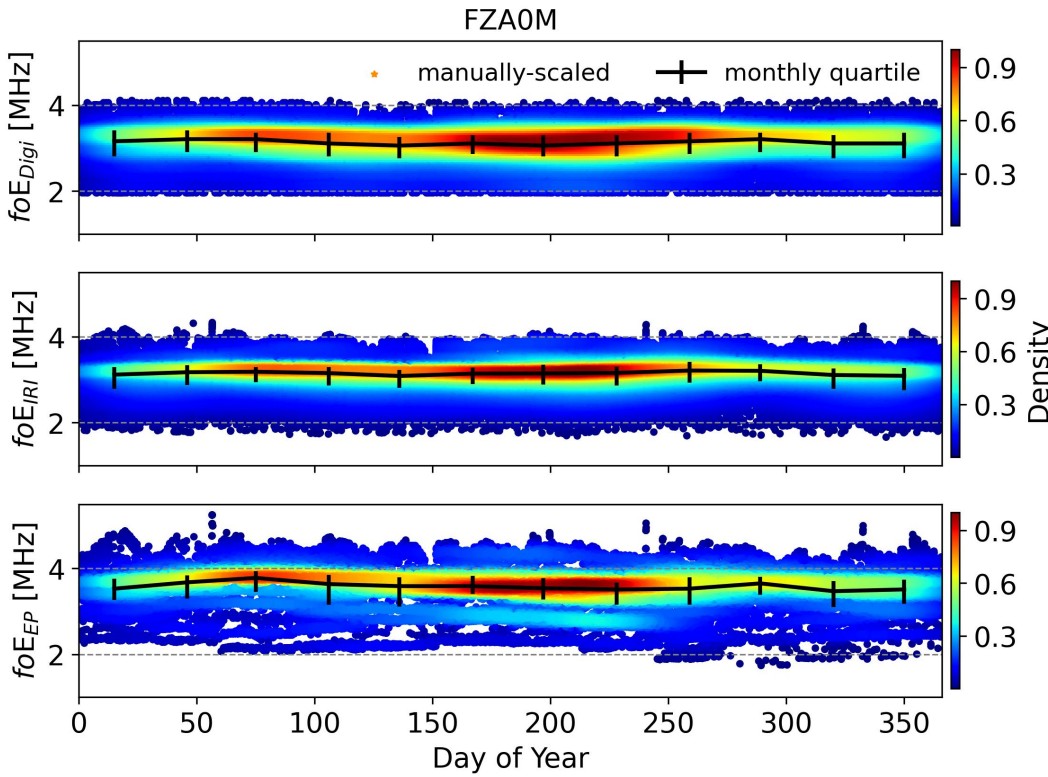

**Figure A5.** Day of Year $fo$E estimates for Fortaleza using ionograms (top), PyIRI (middle), and E-PROBED (bottom). Black trend lines intersect the monthly medians, with quartiles (25% and 75%) shown as error bars.

are present in the ionosonde data with slight decreases in altitude during solar maximum, but these subtle variations are not observed in the model predictions. Interestingly, the few manually-scaled ionograms show a trend similar to that of EA036 where the $hm$E values are much larger than the auto-scaled values.





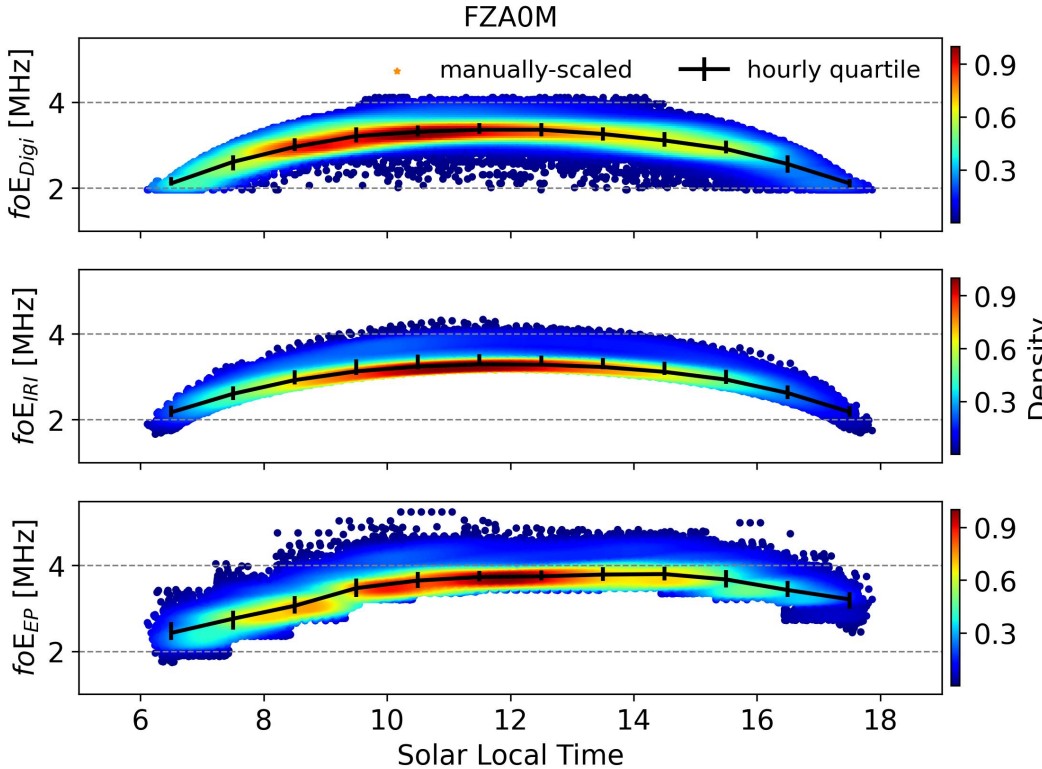

**Figure A6.** Solar local time $foE$ estimates for Fortaleza using ionograms (top), PyIRI (middle), and E-PROBED (bottom). Black trend lines intersect the hourly medians, with quartiles (25% and 75%) shown as error bars.



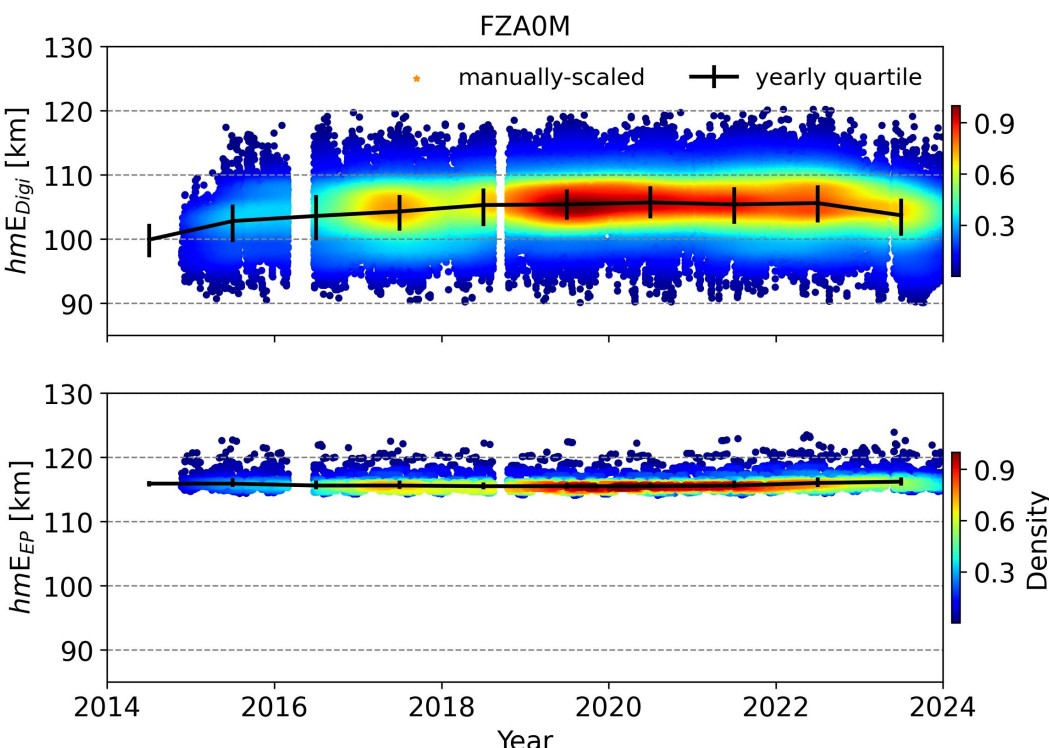

**Figure A7.** Yearly $hm$E estimates for Fortaleza using ionograms (top), and E-PROBED (bottom). Black trend lines intersect the yearly medians, with quartiles (25% and 75%) shown as error bars.



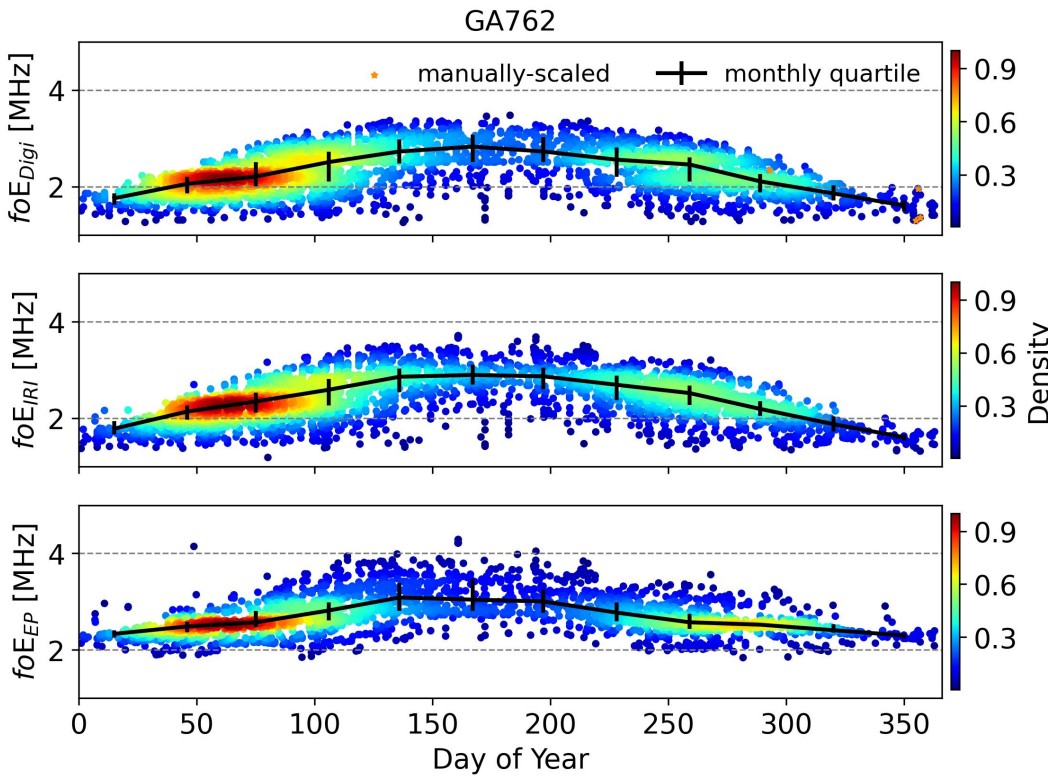

**Figure A8.** Day of Year $foE$ estimates for Gakona using ionograms (top), PyIRI (middle), and E-PROBED (bottom). Black trend lines intersect the monthly medians, with quartiles (25% and 75%) shown as error bars.



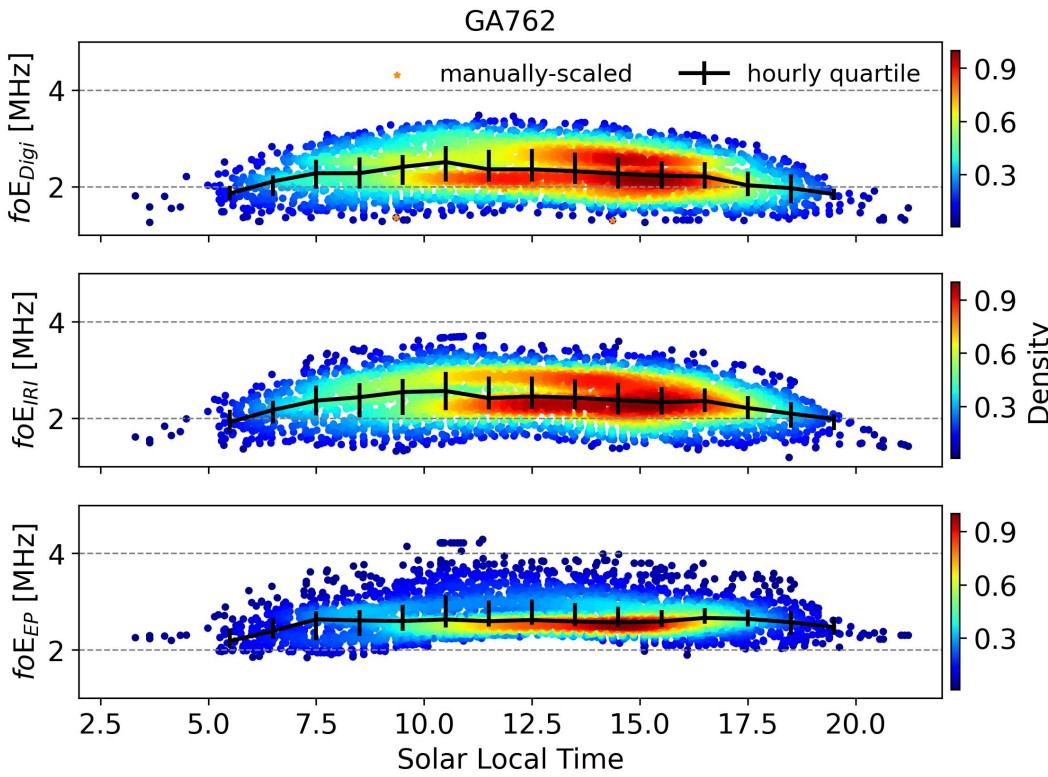

**Figure A9.** Solar local time $foE$ estimates for Gakona using ionograms (top), PyIRI (middle), and E-PROBED (bottom). Black trend lines intersect the hourly medians, with quartiles (25% and 75%) shown as error bars.





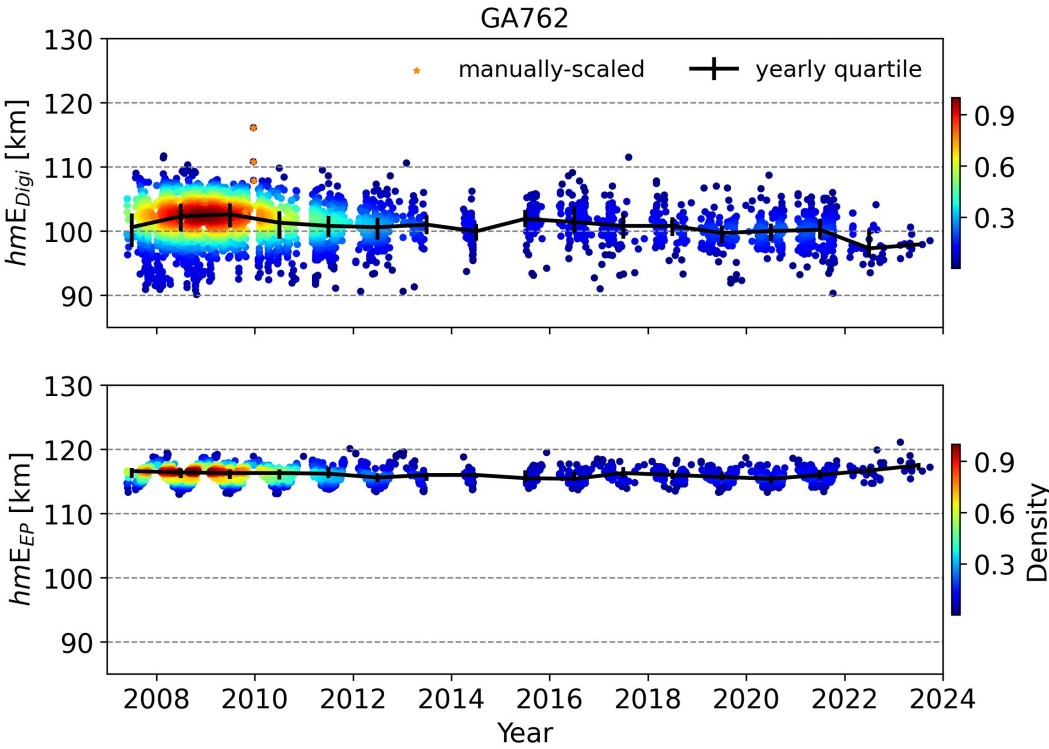

**Figure A10.** Yearly $hm$E estimates for Gakona using ionograms (top) and E-PROBED (bottom). Black trend lines intersect the yearly medians, with quartiles (25% and 75%) shown as error bars.