# Peer review of "A comparison of modeled daytime E-regions from E-PROBED and PyIRI with ionosonde observations"

_EGUsphere, 2025_

## Referee Comment (RC2)

**A comparison of modeled daytime E-regions from E-PROBED and PyIRI with ionosonde observations**

Daniel J. Emmons1, Cornelius Csar Jude H. Salinas2, Dong L. Wu2, Nimalan Swarnalingam2,3, Eugene V. Dao4, Jorge L. Chau5, Yosuke Yamazaki5, Kyle E. Fitch1, and Victoriya V. Forsythe6

1Air Force Institute of Technology, Wright-Patterson AFB, OH, United States 2NASA Goddard Space Flight Center, Greenbelt, MD, United States 3The Catholic University of America, Washington, DC, United States 4Air Force Research Laboratory, Albuquerque, NM, United States 5Leibniz Institute for Atmospheric Physics, Kuhlungsborn, Germany 6U.S. Naval Research Laboratory, Washington, DC, United States Correspondence: Daniel J. Emmons (daniel.emmons@afit.edu)

Thank you for providing the opportunity to review this paper. Please accept it after minor correction.

The authors evaluated two recently developed ionospheric models, PyIRI and E-PROBED, focusing on how accurately they represent the E-region of the ionosphere, which is important for radio wave propagation and ionospheric conductivity. They compared the models' predictions with ionosonde observations collected from three stations located at different latitudes: Fortaleza in Brazil (low latitude), El Arenosillo in Spain (mid latitude), and Gakona in Alaska (high latitude). The comparison covered the years 2009–2024 for El Arenosillo and Gakona, and 2015–2024 for Fortaleza.

The study analyzed two key parameters, foE (the critical frequency of the E-layer) and hmE (the peak height of the E-layer), using both manually scaled and automatically scaled ionograms processed by the ARTIST-5 software. In addition, the paper compared modeled and observed virtual heights using a numerical ray-tracer to evaluate how well the models reproduce altitude-dependent electron density structures.

Their results showed that both models generally agreed well with ionosonde data and successfully captured the solar cycle, seasonal, and diurnal variations of foE. However, E-PROBED tended to overestimate foE, with mean absolute relative errors (MRAE) reaching about 70% at dusk, while PyIRI showed close agreement with observations, with MRAEs around 10%. For hmE, E-PROBED consistently overestimated the height by about 15–20 km compared with auto-scaled ionograms, and PyIRI produced a constant value of 110 km for all times. When compared with manually scaled data, however, E-PROBED's hmE values matched more closely, indicating that auto-scaled data can be less reliable.

Finally, both models produced reasonable virtual height estimates, showing only slight biases relative to ionosonde observations. The direction of the bias differed between manual and auto-scaled datasets, suggesting uncertainties in the automatic scaling process. Overall, the authors concluded that E-PROBED and PyIRI provide reliable and practical representations of the E-region, suitable for applications that require modeled ionospheric parameters.

Here are some minor corrections:

1- In figures displaying **foE** or **hmE** separately for the ionosondes, PyIRI, and E-PROBED (e.g., Figure 1), it would be beneficial to extract the representative **median curves** from each panel and

- present them together in an additional comparative panel. This approach would make direct comparisons clearer and reduce the reader's cognitive effort.
- 2- Please provide a brief explanation of the main differences between **PyIRI** and **E-PROBED** in the introduction or methodology section to help readers understand their respective modeling approaches and assumptions.

---

## Author Response (AR1)

Department of Engineering Physics
Air Force Institute of Technology
2950 Hobson Way
WPAFB Ohio, USA

We sincerely thank you both for your careful reviews and thoughtful feedback; we are grateful for your time and effort. Your reviews have helped refine this manuscript and we address each of your specific comments below.

**Reviewer 1**

- The Mean Relative Absolute Error (MRAE) values, for instance in Figure 2, are currently presented on a logarithmic scale. While such representation emphasizes the dynamic range, a linear scale would provide a more direct appreciation of error magnitudes and may be more accessible to the general readership. This reviewer recommends the authors to consider revising the figure accordingly
  - We originally used the logarithmic scale such that a single colorbar could be used for both the PyIRI and E-PROBED MRAE subfigures. However, we agree that a linear scale would be more accessible for readers, and we have changed the scale for each of the MRAE figures from logarithmic to linear. A separate colorbar is now used for each subfigure.
- A salient result is the significant offset between E-PROBED and ionosonde foE near dusk. This reviewer suggests the authors to provide additional physical interpretation of this phenomenon. Possible contributing factors may include ionospheric tilts, GNSS-RO retrieval geometry, or specific assumptions intrinsic to the model. A brief discussion would provide valuable context
  - This significant offset is certainly worthy of additional discussion in the manuscript. As you suggest, ionospheric tilts and other causes of horizontal density gradients near dusk such as the Appleton Anomaly with pre-reversal enhancement can impact GNSS-RO derived electron density profile estimates. As shown in Wu et al., (2023), high inclination (cross-latitude) sensors such as COSMIC-1 are more sensitive to ionospheric inhomogeneities. Since E-PROBED was derived from COSMIC-1 observations, and there are known horizontal density gradients during dusk, it stands to reason that the RO derived EDPs are likely impacted by the inhomogeneities, resulting in foE overestimates by E-PROBED near dusk. This explanation has been added to the paragraph discussing ionospheric tilts in the Discussion section.
- A central theme of the manuscript concerns the discrepancy between auto-scaled and manually scaled ionograms. In some instances (e.g., EA036 in 2009), the divergence is substantial not only relative to manually scaled values but also with respect to the modal behavior of the dataset. This raises a fundamental question regarding data reliability: in cases of large offsets, which dataset should be deemed more trustworthy? If the manually

scaled values are regarded as the reference truth, the validity of long-term comparisons may be undermined, given the sparse availability of such data, largely limited to 2009. The authors acknowledge this limitation, but a more explicit statement on how future studies might reconcile these inconsistencies would strengthen the manuscript

- o We completely agree that the discrepancy between manual and auto-scaled ionograms is problematic for comparing long-term hmE trends. While the foE trends between manual and auto-scaled ionograms agree, the large differences in hmE trends caused us to move the majority of the hmE figures to the Appendix along with a caution that the results must be taken with great care. Ideally, a future study with a long-term collection of manually-scaled ionograms could be performed to reanalyze the hmE trends and remove these inconsistencies. We have added explicit statements on this topic to Section 3.1 around Figure 3 as well as Section 4 (Discussion).

- In figures where foE or hmE are displayed separately for ionosondes, PyIRI, and E-PROBED (e.g., Figure 1), it would be advantageous to extract the representative median curves from each panel and combine them into an additional comparative panel. Such an approach would greatly facilitate direct comparison and reduce the cognitive load on the reader

  - o Thank you for the suggestion, this addition certainly makes the comparison easier for readers. We added the median curves for each of the trends (ionosonde, PyIRI, and E-PROBED) to each of the subfigures for the yearly, seasonal, and diurnal figures. The two reference trends for each subfigure are semi-transparent but visible to compare against the primary dataset and trend.

- While the manuscript adequately presents the numerical performance of both models, additional emphasis on their conceptual distinctions, namely, PyIRI as an ionosonde-driven semi-empirical model versus E-PROBED as a GNSS-RO-based climatological model, would better contextualize the observed biases and delineate the respective domains of applicability. In addition, the essential differences between PyIRI and the conventional IRI (Fortran) remain insufficiently explained. Since many readers may not be familiar with these distinctions, this reviewer suggests the authors to provide a more detailed introduction to both models

  - o Thank you for pointing out this omission. We have added a more detailed introduction to both models in the Introduction section, including details on the differences between PyIRI and IRI. Some of the E-PROBED introduction was moved from the Methodology section and placed in the Introduction section to improve the flow of the document.

- The comparison of results across low-, mid-, and high-latitude stations is an important strength of this work. Nevertheless, further discussion on latitude-dependent ionospheric drivers would affect the results. For example, auroral electron precipitation at high latitudes, sporadic E contamination at mid latitudes, and the equatorial electrojet at low latitudes may differentially affect the observed discrepancies between E-PROBED and ionosondes. Furthermore, GNSS-RO retrieval geometry differs across latitudes, since most GNSS satellites are not polar orbiting, potentially reducing accuracy at higher latitudes

  - o The contamination of both the ionosonde and GNSS-RO observations caused by ionospheric irregularities is certainly worthy of discussion as the various

irregularities will produce differential uncertainties in the datasets for the different ionosonde sites and measurement techniques. The GNSS-RO retrieval geometry also increases the likelihood of encountering ionospheric irregularities due to the integrated nature of the observations that traverse large horizontal distances. We added a paragraph in the Discussion section to address these issues.

**Reviewer 2**

- In figures displaying foE or hmE separately for the ionosondes, PyIRI, and E-PROBED (e.g., Figure 1), it would be beneficial to extract the representative median curves from each panel and present them together in an additional comparative panel. This approach would make direct comparisons clearer and reduce the reader's cognitive effort.

    o Thank you for the suggestion, this addition will certainly make the comparisons clearer and reduce the readers' cognitive efforts. We added the median curves for each of the trends (ionosonde, PyIRI, and E-PROBED) to each of the other subfigures for easier comparison. The reference trends are semi-transparent but visible for comparison against the primary trend for each subfigure.

- Please provide a brief explanation of the main differences between PyIRI and E-PROBED in the introduction or methodology section to help readers understand their respective modeling approaches and assumptions.

    o Thank you for catching this omission in the manuscript. We added a brief explanation of the main differences and drivers for PyIRI and E-PROBED in the Introduction Section, and some of the E-PROBED model description was moved from the Methodology Section to the Introduction Section to improve the flow of the document.

Once again, we thank the referees for their thoughtful and detailed reviews.

Daniel Emmons

daniel.emmons@afit.edu